# Consistent Flow Distillation for Text-to-3D Generation

**Runjie Yan**[*]
UC San Diego

**Yinbo Chen**[*]
UC San Diego

**Xiaolong Wang**
UC San Diego

## Abstract

Score Distillation Sampling (SDS) has made significant strides in distilling image-generative models for 3D generation. However, its maximum-likelihood-seeking behavior often leads to degraded visual quality and diversity, limiting its effectiveness in 3D applications. In this work, we propose Consistent Flow Distillation (CFD), which addresses these limitations. We begin by leveraging the gradient of the diffusion ODE or SDE sampling process to guide the 3D generation. From the gradient-based sampling perspective, we find that the consistency of 2D image flows across different viewpoints is important for high-quality 3D generation. To achieve this, we introduce multi-view consistent Gaussian noise on the 3D object, which can be rendered from various viewpoints to compute the flow gradient. Our experiments demonstrate that CFD, through consistent flows, significantly outperforms previous methods in text-to-3D generation. Project page: https://runjie-yan.github.io/cfd/.

## 1 Introduction

3D content generation has been gaining increasing attention in recent years for its wide range of applications. However, it is expensive to create high-quality 3D assets or scan objects in the real world. The scarcity of 3D data has been a primary challenge in 3D generation. On the other hand, image synthesis has witnessed great progress, particularly with diffusion models trained on large-scale datasets with massive high-quality and diverse images. Leveraging the 2D generative knowledge for 3D generation by model distillation has become a research direction of key importance.

Score Distillation Sampling (Poole et al., 2023) (SDS) pioneered the paradigm. It uses a pretrained text-to-image diffusion model to optimize a single 3D representation such that the rendered views seek a maximum likelihood objective. Several subsequent efforts (Zhu et al., 2024; Liang et al., 2023; Katzir et al., 2024; Huang et al., 2024; Tang et al., 2023; Wang et al., 2023b; Armandpour et al., 2023) have been made to improve SDS, while the maximum-likelihood-seeking behavior remains, which has a detrimental effect on the visual quality and diversity. Variational Score Distillation (Wang et al., 2024a) (VSD) tackles this issue by treating the 3D representation as a random variable instead of a single point as in SDS. However, the random variable is simulated by particles in VSD. Single-particle VSD is theoretically equivalent to SDS (Wang et al., 2023b), assuming the LoRA network in VSD is always trained to optimal. While the optimization-based sampling of VSD is $k$ times slower with $k$ particles.

In this work, we propose Consistent Flow Distillation (CFD), which distills 3D representations through gradient-based diffusion sampling of consistent 2D image probability flows across different views. We provide theoretical analysis of this process and extend it to a wide range of deterministic and stochastic diffusion sampling processes. In the distillation process, we identify that a key is to apply consistent flows to the 3D representation. Intuitively, in 2D image generation, the same region is always associated with the same fixed noise for the correct flow sampling. Analogously, in 3D generation, the 2D image flows from different camera views should also use the noise patterns that are consistent on the object surface with correct correspondence. To achieve this, we design a *multi-view consistent Gaussian noise* based on Noise Transport Equation (Chang et al., 2024), which can compute the multi-view consistent noise with negligible cost. During the distillation process, the

---

[*]Equal contribution

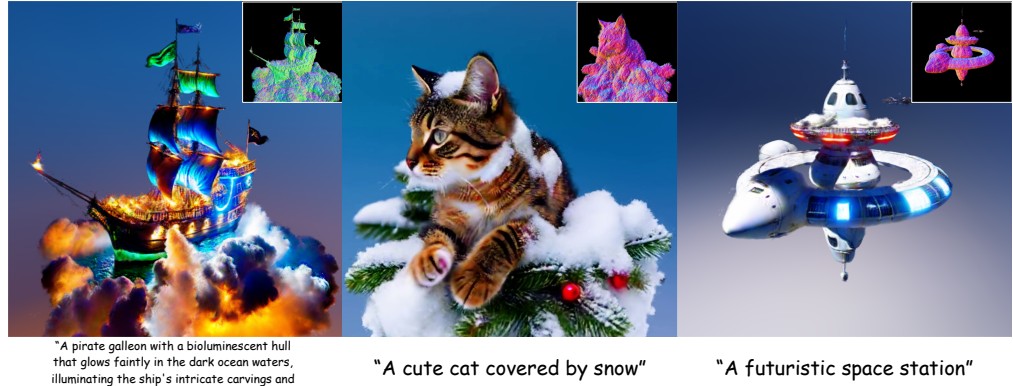

"A pirate galleon with a bioluminescent hull that glows faintly in the dark ocean waters, illuminating the ship's intricate carvings and sails as it silently navigates the waves"

"A cute cat covered by snow"

"A futuristic space station"

(a) NeRFs generated by CFD from scratch.

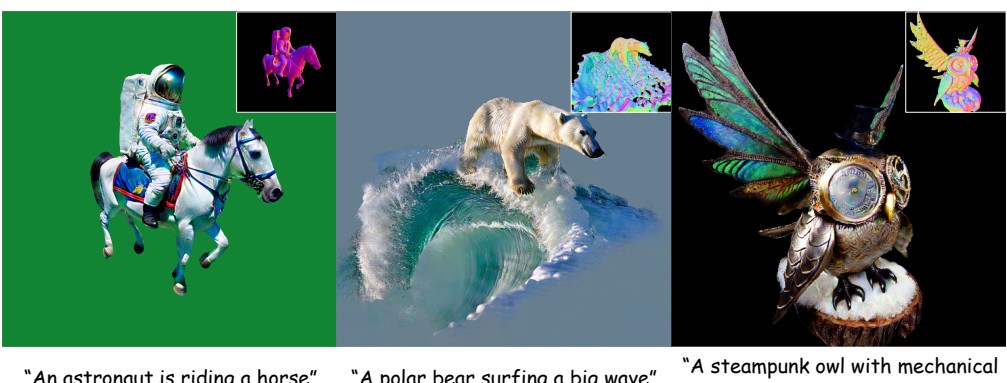

"An astronaut is riding a horse"

"A polar bear surfing a big wave"

"A steampunk owl with mechanical wings"

(b) 3D textured meshes generated by CFD from scratch.

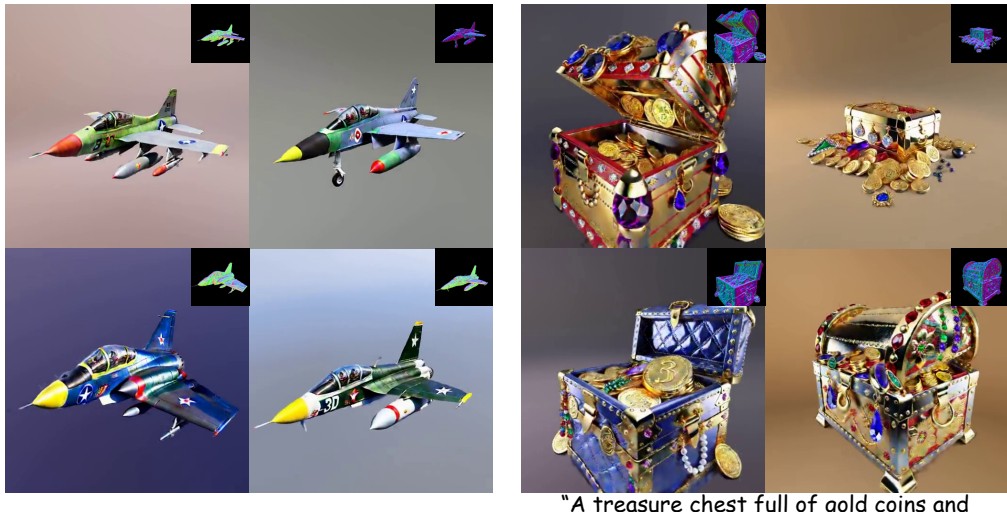

"A 3D model of a toy fighter plane, sharp"

"A treasure chest full of gold coins and jewels, high resolution, sharp"

(c) CFD can generate diverse and high-quality 3D samples from scratch.

Figure 1: **Text-to-3D samples of CFD.** CFD can generate diverse 3D samples by distilling text-to-image diffusion models. See videos in our project page for additional generation results.

multi-view consistent Gaussian noise is rendered from different views to compute the gradient of 2D image flow. Finally, our method can create high quality and diverse 3D objects by following the diffusion ODE or SDE sampling process.

We evaluate our method with different types of pretrained 2D image diffusion models, and compare it with state-of-the-art text-to-3D score distillation methods. Both qualitative and quantitative experiments show the effectiveness of our approach compared with prior works. Our method generates 3D assets with realistic appearance and shape (Fig. 1(a), 1(b)) and can sample diverse 3D objects for the same text prompt (Fig. 1(c)) with negligible extra computation cost compared with SDS.

In summary, our main contributions are:

- An in-depth discussion about using image diffusion PF-ODE or SDE to directly guide 3D generation. We present equivalent forms of the ODE and SDE so that their random variables are clean images at any time in the diffusion process, and identified that flow consistency is a key in this process.

- A multi-view consistent Gaussian noise on the 3D object, that keeps pixel i.i.d. Gaussian property in any single view and has correct correspondence on the object surface between different views.

- A method to distill image diffusion models for 3D generation. It is as simple and efficient as SDS while having significantly better quality and diversity.

## 2 PRELIMINARIES

### 2.1 DIFFUSION MODELS AND PROBABILITY FLOW ORDINARY DIFFERENTIAL EQUATION (PF-ODE)

A forward diffusion process (Sohl-Dickstein et al., 2015; Ho et al., 2020) gradually adds noise to a data point $\boldsymbol{x}_0 \sim p_0(\boldsymbol{x}_0)$, such that the intermediate distribution $p_{t0}(\boldsymbol{x}_t|\boldsymbol{x}_0)$ conditioned on initial sample $\boldsymbol{x}_0$ at diffusion timestep $t$ is $\mathcal{N}(\alpha_t\boldsymbol{x}_0, \sigma_t^2\boldsymbol{I})$, which can be equivalently written as

$$\boldsymbol{x}_t = \alpha_t\boldsymbol{x}_0 + \sigma_t\boldsymbol{\epsilon}, \quad \boldsymbol{\epsilon} \sim \mathcal{N}(\boldsymbol{0}, \boldsymbol{I}), \tag{1}$$

where $\alpha_0 = 1, \sigma_0 = 0$ at the beginning, and $\alpha_T \approx 0, \sigma_T \approx 1$ in the end, such that $p_T(\boldsymbol{x}_T)$ is approximately the standard Gaussian $\mathcal{N}(\boldsymbol{0}, \sigma_T^2\boldsymbol{I})$. A diffusion model $\boldsymbol{\epsilon}_\phi$ is learned to reverse such process, typically with the following denoising training objective (Ho et al., 2020):

$$\mathcal{L}_{\mathrm{DM}}(\phi) = \mathbb{E}_{\boldsymbol{x}_0, \boldsymbol{\epsilon}, t}[w_t||\boldsymbol{\epsilon}_\phi(\boldsymbol{x}_t, t) - \boldsymbol{\epsilon}||_2^2]. \tag{2}$$

After training, $\boldsymbol{\epsilon}_\phi(\boldsymbol{x}_t, t) \approx -\sigma_t\nabla_{\boldsymbol{x}_t}\log p_t(\boldsymbol{x}_t)$, where $\nabla_{\boldsymbol{x}_t}\log p_t(\boldsymbol{x}_t)$ is termed *score function*.

A Probability Flow Ordinary Differential Equation (PF-ODE) has the same marginal distribution as the forward diffusion process at any time $t$ (Song et al., 2021b). The PF-ODE can be written as:

$$\frac{\mathrm{d}(\boldsymbol{x}_t/\alpha_t)}{\mathrm{d}t} = \frac{\mathrm{d}(\sigma_t/\alpha_t)}{\mathrm{d}t}\left(-\sigma_t\nabla_{\boldsymbol{x}_t}\log p_t(\boldsymbol{x}_t)\right) \tag{3}$$

$$= \frac{\mathrm{d}(\sigma_t/\alpha_t)}{\mathrm{d}t}\boldsymbol{\epsilon}_\phi(\boldsymbol{x}_t, t), \quad \boldsymbol{x}_T \sim p_T(\boldsymbol{x}_T). \tag{4}$$

A data point $\boldsymbol{x}_0$ can be sampled by starting from a Gaussian noise $\boldsymbol{x}_T \sim \mathcal{N}(\boldsymbol{0}, \sigma_T^2\boldsymbol{I})$ and following the PF-ODE trajectory from $t = T$ to $t = 0$, typically with discretized timesteps and an ODE solver.

### 2.2 DIFFERENTIABLE 3D REPRESENTATIONS

Differentiable 3D representations are typically parameterized by the learnable parameters $\theta$ and a differentiable rendering function $\boldsymbol{g}_\theta(\boldsymbol{c})$ to render images corresponding to the camera views $\boldsymbol{c}$. In many tasks, the gradient is first obtained on the rendered images $\boldsymbol{g}_\theta(\boldsymbol{c})$ and then backpropagated through the Jacobian matrix $\frac{\partial\boldsymbol{g}_\theta(\boldsymbol{c})}{\partial\theta}$ of the renderer to the learnable parameters $\theta$.

Common 3D neural representations include Neural Radiance Field (NeRF) (Mildenhall et al., 2021; Müller et al., 2022; Wang et al., 2021; Barron et al., 2021; Xu et al., 2022), 3D Gaussian Splatting

(3DGS) (Kerbl et al., 2023), and Mesh (Laine et al., 2020; Shen et al., 2021). In this work, we perform experiments on various 3D representations and validate that our method is applicable for generation across a wide range of 3D representations.

## 3 CONSISTENT FLOW DISTILLATION

We present Consistent Flow Distillation (CFD), which takes a pretrained and frozen text-to-image diffusion model and distills a 3D representation by the gradient from the probability flow of the 2D image diffusion model. We propose to guide 3D generation with 2D clean flow gradients operating jointly on a 3D object. We identify that a key in this process is to make the flow guidance consistent across different camera views (see Sec. 3.1). We further propose an SDE, a generalization of the clean flow ODE, that incorporates noise injection during optimization to enhance generation quality (see Sec. 3.2). To achieve the consistent flow, we propose an algorithm to compute a multi-view consistent Gaussian noise, which provides noise for different views with noise texture exactly aligned on the surface of the 3D object (see Sec. 3.3). Finally, we draw connections between CFD and other score distillation methods (see Sec. 3.4).

### 3.1 3D GENERATION WITH 2D CLEAN FLOW GRADIENT

Given a pretrained text-to-image diffusion model $\boldsymbol{\epsilon}_\phi(\boldsymbol{x}_t, t, y)$, let $y$ denote the condition (text prompt), the conditional distribution $p(\boldsymbol{x}_0|y)$ can be sampled from the PF-ODE (Song et al., 2021b) trajectory from $t = T$ to $t = 0$, which takes the form

$$\mathrm{d}(\frac{\boldsymbol{x}_t}{\alpha_t}) = \underbrace{\mathrm{d}(\frac{\sigma_t}{\alpha_t})}_{-lr} \cdot \underbrace{\boldsymbol{\epsilon}_\phi(\boldsymbol{x}_t, t, y)}_{\nabla\mathcal{L}}. \tag{5}$$

By following the diffusion PF-ODE, pure Gaussian noise is transformed to an image in the target distribution $p(\boldsymbol{x}_0|y)$. Thus PF-ODE can be interpreted as guiding the refinement of a noisy image to a realistic image. Can we use image PF-ODE to directly guide the generation of a differentiable 3D representation $\theta$ through the refining process, with $\theta$ as its learnable parameters and $\boldsymbol{g}_\theta$ as its differentiable rendering function?

A direct implementation can be substituting the noisy images in Eq. 5 with the rendered images $\boldsymbol{g}_\theta(\boldsymbol{c})$ at the camera view $\boldsymbol{c}$ by letting $\frac{\boldsymbol{x}_t}{\alpha_t} = \boldsymbol{g}_\theta(\boldsymbol{c})$. By viewing $\mathrm{d}(\frac{\sigma_t}{\alpha_t})$ as the learning rate $lr$ of an optimizer and $\boldsymbol{\epsilon}_\phi(\boldsymbol{x}_t, t, y)$ as the loss gradient to $\frac{\boldsymbol{x}_t}{\alpha_t}$, the gradient can be backpropagated through the Jacobian matrix of the renderer $\boldsymbol{g}_\theta(\boldsymbol{c})$ to update $\theta$ according to

$$\Delta\theta = -lr \cdot \boldsymbol{\epsilon}_\phi(\alpha_t \boldsymbol{g}_\theta(\boldsymbol{c}), t, y)\frac{\partial \boldsymbol{g}_\theta(\boldsymbol{c})}{\partial\theta}. \tag{6}$$

However, such a direct attempt may not work (see Fig. 5 (a)), since the image $\boldsymbol{x}_t$ at diffusion timestep $t$ contains Gaussian noise. It is hard for the images rendered by a 3D representation to match the noisy images $\frac{\boldsymbol{x}_t}{\alpha_t}$ in an image PF-ODE, particularly around the beginning $t = T$, where $\boldsymbol{x}_T$ is per-pixel independent Gaussian noise. It is generally impossible for a continuous 3D representation to be rendered as per-pixel independent Gaussian noise from all camera views simultaneously. As a result, the rendered views may be out-of-distribution (OOD) as the input to the pretrained image diffusion model, and therefore cannot get meaningful gradient as guidance.

To resolve the OOD issue, we use a *change-of-variable* (Gu et al., 2023; Yan et al., 2024) to transform the original *noisy variable* $\boldsymbol{x}_t$ in PF-ODE (Eq. 5) to a new variable that is free of Gaussian noise at any time $t \in [0, T]$. For each trajectory $\{\boldsymbol{x}_t\}_{t\in[0,T]}$ of the $\boldsymbol{x}_t$ in the original PF-ODE, the new variable $\hat{\boldsymbol{x}}_t^c$ is defined as

$$\hat{\boldsymbol{x}}_t^c \triangleq \frac{\boldsymbol{x}_t - \sigma_t\tilde{\boldsymbol{\epsilon}}}{\alpha_t}, \tag{7}$$

where $\tilde{\boldsymbol{\epsilon}}$ is set as the initial noise $\tilde{\boldsymbol{\epsilon}} = \frac{\boldsymbol{x}_T}{\sigma_T}$ and is a constant for each ODE trajectory $\{\boldsymbol{x}_t\}_{t\in[0,T]}$. By Eq. 5 and Eq. 7, the evolution of the new variable $\hat{\boldsymbol{x}}_t^c$ is derived as follows:

$$\mathrm{d}\hat{\boldsymbol{x}}_t^c = \underbrace{\mathrm{d}(\frac{\sigma_t}{\alpha_t})}_{-lr} \cdot \underbrace{\left(\boldsymbol{\epsilon}_\phi(\alpha_t\hat{\boldsymbol{x}}_t^c + \sigma_t\tilde{\boldsymbol{\epsilon}}, t, y) - \tilde{\boldsymbol{\epsilon}}\right)}_{\nabla\mathcal{L}}. \tag{8}$$

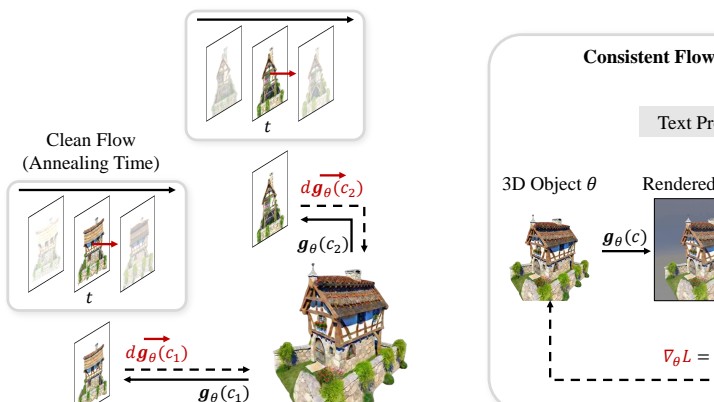

Figure 2: **Overview of CFD.** The 3D representation $\theta$ is generated with decreasing timesteps. At each timestep $t$, different views $g_\theta(c)$ are rendered. The 2D image clean flow provides the gradient at timestep $t$ to the views and backpropagates to $\theta$. The right shows the gradient computation in detail: we add a multi-view consistent noise (see Fig. 3) to the rendered image and pass it into the frozen text-to-image diffusion model, gradient is calculated using the model prediction and then backpropagated to $\theta$.

Changing the variable $\boldsymbol{x}_t$ of the original diffusion PF-ODE to the variable $\hat{\boldsymbol{x}}_t^c$ makes directly using PF-ODE as a 3D guidance possible by providing the following properties (the proof is in Appx. G.3): (i) $\hat{\boldsymbol{x}}_t^c$ are clean images for all $t \in [0, T]$ (see Appx. Fig. 17), therefore, it can be substituted with the rendered clean images $g_\theta(\boldsymbol{c})$. (ii) $\hat{\boldsymbol{x}}_t^c$ is initialized from zero: $\hat{\boldsymbol{x}}_T^c = \boldsymbol{0}$, which can be consistent with the 3D representation initialization (e.g. NeRF, where the entire scene is initialized to a uniform gray). (iii) The endpoint of the new ODE trajectory $\hat{\boldsymbol{x}}_0^c = \boldsymbol{x}_0$ is a sample following the target distribution $p_0(\boldsymbol{x}_0)$ and is completely determined by the constant $\tilde{\boldsymbol{\epsilon}}$ (thus $\tilde{\boldsymbol{\epsilon}}$ can be viewed as the identity of the trajectory). The new variable $\hat{\boldsymbol{x}}_t^c$ is therefore termed *clean variable*. Note that $\hat{\boldsymbol{x}}_t^c$ is different from the "*sample prediction*" $\hat{\boldsymbol{x}}_t^{\text{gt}} \triangleq \frac{\boldsymbol{x}_t - \sigma_t \boldsymbol{\epsilon}_\phi(\boldsymbol{x}_t, t, y)}{\alpha_t}$ of diffusion network for $\boldsymbol{x}_t$, which is not directly usable in this framework and we discuss for more details in Appx. H. We use *clean flow* to denote the ODE (Eq. 8) of the clean variable $\hat{\boldsymbol{x}}_t^c$.

Similar to Eq. 6, we use the following gradient to update the 3D representation $\theta$:

$$\nabla_\theta \mathcal{L}_{\text{CFD}}(\theta) = \mathbb{E}_c \left[ \left( \boldsymbol{\epsilon}_\phi(\alpha_t \boldsymbol{g}_\theta(\boldsymbol{c}) + \sigma_t \tilde{\boldsymbol{\epsilon}}(\theta, \boldsymbol{c}), t, y) - \tilde{\boldsymbol{\epsilon}}(\theta, \boldsymbol{c}) \right) \frac{\partial \boldsymbol{g}_\theta(\boldsymbol{c})}{\partial \theta} \right], \tag{9}$$

where $t = t(\tau)$ is a predefined monotonically decreasing timestep annealing function of the optimization time $\tau$, and $\tilde{\boldsymbol{\epsilon}}(\theta, \boldsymbol{c})$ is a multi-view consistent Gaussian noise function, we discuss its design details in Sec. 3.3. We let $\tilde{\boldsymbol{\epsilon}}(\theta, \boldsymbol{c})$ be a deterministic function of $\theta$ and $c$, ensuring that the noise remains constant for a fixed camera view and geometry, given that $\tilde{\boldsymbol{\epsilon}}$ is constant for a single flow trajectory in clean flow ODE. Since we have a set of 2D image flows jointly operating on a 3D object, the gradient updates from different camera views in Eq. 9 may interfere with each other. We identify that a key in the 3D sampling process is to make the 2D image flows consistent on the 3D object surface. This requires a multi-view consistent Gaussian noise function $\tilde{\boldsymbol{\epsilon}}(\theta, \boldsymbol{c})$ that is not only view-dependent but also provides the correct local correlation on the object surface. The multi-view consistent Gaussian noise function should apply a similar noise pattern to the same region of the object surface, even from different camera views. This corresponds to that the fixed noise pattern is always added to the same region for the clean variable in 2D image clean flow ODE. The overall process of CFD is summarized in Fig. 2.

## 3.2 GUIDING 3D GENERATION WITH DIFFUSION SDE

Despite that PF-ODE and diffusion SDE can recover the same marginal distributions in theory, SDE-based stochastic sampling may result in better generation quality as reported in prior works (Song

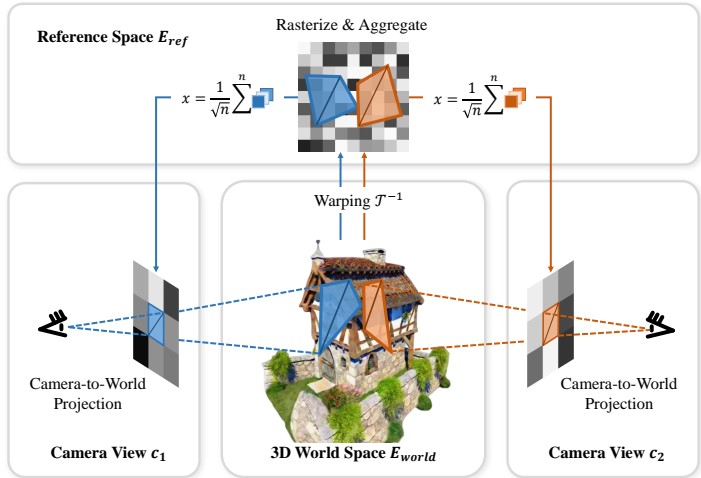

Figure 3: **Warping consistent noise for query views.** To obtain a query view noise map, for each pixel, its vertices are projected onto the object surface, then wrapped to the coordinates in a high-resolution noise map. The values within the region specified by the coordinates on the high-resolution noise map are summed and normalized as the return pixel value in the query view noise.

et al., 2021b;a; Karras et al., 2022). Motivated by this, we also propose to use image diffusion SDE to guide 3D generation.

To achieve this, we propose a reverse-time SDE with a form similar to the clean flow ODE (Eq. 8):

$$
\begin{cases}
\mathrm{d}\hat{\boldsymbol{x}}_t^{\mathrm{c}} = \underbrace{\left(\mathrm{d}(\frac{\sigma_t}{\alpha_t}) + \frac{\sigma_t}{\alpha_t}\beta_t\mathrm{d}t\right)}_{-lr} \cdot \underbrace{\left(\boldsymbol{\epsilon}_\phi(\alpha_t\hat{\boldsymbol{x}}_t^{\mathrm{c}} + \sigma_t\tilde{\boldsymbol{\epsilon}}_t, t, y) - \tilde{\boldsymbol{\epsilon}}_t\right)}_{\nabla\mathcal{L}}, \\
\mathrm{d}\tilde{\boldsymbol{\epsilon}}_t = \tilde{\boldsymbol{\epsilon}}_t\beta_t\mathrm{d}t + \sqrt{2\beta_t}\mathrm{d}\bar{\boldsymbol{w}}_t,
\end{cases}
\tag{10}
$$

with initial condition $\hat{\boldsymbol{x}}_T^{\mathrm{c}} = \boldsymbol{0}$ and $\tilde{\epsilon}_T \sim \mathcal{N}(\boldsymbol{0}, \boldsymbol{I})$, where $\bar{\boldsymbol{w}}_t$ is a standard Wiener process in the reverse time from $T$ to $0$. It can be further proved that this SDE and its forward-time form are equivalent to the diffusion SDE presented by Song et al. (Song et al., 2021b) and EDM (Karras et al., 2022). When we set $\beta_t = 0$, the SDE becomes deterministic and becomes the clean flow ODE. When $\beta_t \neq 0$, new Gaussian noise will be injected into $\tilde{\epsilon}_t$ during the diffusion process, but $\tilde{\epsilon}_t$ is still of unit variance throughout the whole process from $T$ to $0$. Furthermore, $\hat{\boldsymbol{x}}_t^{\mathrm{c}}$ in this SDE still retains the "clean properties" of $\hat{\boldsymbol{x}}_t^{\mathrm{c}}$ in the clean flow ODE. Thus, we also use clean flow to refer to this SDE. We provide detailed discussions and proofs about this SDE in Appx. G

The clean flow SDE implies that a simple modification on Eq. 9 can make $\nabla_\theta\mathcal{L}_{\mathrm{CFD}}(\theta)$ correspond to using SDE guidance. As detailed in Appx. G.4.1, we only need to inject new Gaussian noise into $\tilde{\epsilon}(\theta, \boldsymbol{c})$ during optimization by:

$$
\tilde{\epsilon}(\tau + 1) = \sqrt{1 - \gamma}\tilde{\epsilon}(\tau) + \sqrt{\gamma}\boldsymbol{\epsilon},
\tag{11}
$$

where $\gamma$ is a predefined noise injection rate, $\tau$ is the optimization step, and $\boldsymbol{\epsilon} \sim \mathcal{N}(\boldsymbol{0}, \boldsymbol{I})$ is sampled at each optimization step.

### 3.3 MULTI-VIEW CONSISTENT GAUSSIAN NOISE $\tilde{\epsilon}$

To get consistent flow, a multi-view consistent Gaussian noise function $\tilde{\epsilon}(\theta, \boldsymbol{c})$ is required, which (i) is a per-pixel independent Gaussian noise for all camera views $\boldsymbol{c}$; (ii) the noise patterns from different views have the correct correspondence according to the 3D object surface. It is non-trivial to satisfy all these properties with common warping and interpolation methods. The query rays from camera views $\boldsymbol{c}$ take continuous coordinates, simply using common interpolation methods such as bilinear may break the per-pixel independent property and result in bad quality (see Fig. 5 (b)).

Inspired by Integral Noise (Chang et al., 2024), we develop an algorithm that implements the multi-view consistent Gaussian noise with Noise Transport Equation. The Noise Transport Equation was originally proposed for warping noise between two frames in a video (Chang et al., 2024). To use it in the 3D task, we generalize the Noise Transport Equation to the warping between two different manifolds and compute the warping from different query camera views to the same reference space $E_{ref}$. As shown in Fig. 3, given a camera view $c$, the query pixel $p$ is first projected onto the surface of the object as camera-to-world $ctw_c(p)$ in the world space $E_{world}$, then we map those points from the surface to a reference space $E_{ref}$ through a predefined mapping function $\mathcal{T}^{-1}$ (design details are in Appx. D). We define a high-resolution Gaussian noise map $W$ on $E_{ref}$. Finally, we aggregate and return the noise value $G(p)$ for the query pixel $p$ according to

$$G(p) = \frac{1}{\sqrt{|\Omega_p|}} \sum_{A_i \in \Omega_p} W(A_i),$$ (12)

where $\Omega_p = \mathcal{T}^{-1}(ctw_c(p))$ is the area covered by $p$ after $p$ being warped to $E_{ref}$, $A_i$ is a noise cell in $E_{ref}$, and $W(A_i)$ is the noise value of unit variance at $A_i$. By first projecting query pixels from different camera views to the object surface in the world space $E_{world}$, two query pixels $p_1, p_2$ from two different camera views that look at the same region on the object will be projected to overlapped regions $ctw_{c_1}(p_1)$, $ctw_{c_2}(p_2)$ on the object. After being warped by the same function $\mathcal{T}^{-1}$, they cover overlapped regions $\Omega_{p_1}$, $\Omega_{p_2}$ and get correct correlation in noise maps $G(p_1)$, $G(p_2)$.

Our method can be also viewed as deriving a rendering function for the noisy variable $x_t$ in the original form of PF-ODE (Eq. 5) by

$$x_t(\theta, c) = \alpha_t g_\theta(c) + \sigma_t \tilde{\epsilon}(\theta, c).$$ (13)

As discussed in Integral Noise (Chang et al., 2024), the warping of an image $g_\theta(c)$ follows the transport equation that takes a similar form of Eq. 12, but with a different denominator $|\Omega_p|$, instead of $\sqrt{|\Omega_p|}$ for $\tilde{\epsilon}(\theta, c)$, thus common 3D representation is incapable of rendering Gaussian Noise $\tilde{\epsilon}(\theta, c)$, and it is needed to disentangle the noisy variable into the clean part $g_\theta(c)$ and noisy part $\tilde{\epsilon}(\theta, c)$. By disentanglement, we can handle the two parts that follow different rendering equations separately and achieve the rendering of the noisy variable for using image PF-ODE (or diffusion SDE) as the guidance for 3D generation.

### 3.4 COMPARISON WITH OTHER SCORE DISTILLATION METHODS

**Comparison with SDS.** Both SDS and our CFD share a similar gradient form $\left(\epsilon_\phi(x_t, t, y) - \tilde{\epsilon}\right) \frac{\partial g_\theta(c)}{\partial \theta}$ to update the 3D representation $\theta$ from a sampled rendered view. In SDS, $t$ is typically randomly sampled from a range $[t_{\min}, t_{\max}]$, and $\tilde{\epsilon}$ is a noise randomly sampled at each step. In contrast to SDS, our CFD uses an annealing timestep $t(\tau)$ that decreases from $t_{\max}$ to $t_{\min}$, the deterministic noise $\tilde{\epsilon}(\theta, c)$ depends on both the object surface and the camera view, it is designed to let the noise from different views have correct correspondence according to the object surface. Notably, SDS with annealing timestep schedule can be viewed as setting $\gamma = 1$ in CFD, where significant stochasticity is injected in the optimization. As a comparison, for typical diffusion sampling processes, $\gamma \approx 0.00024$ in DDPM, and $\gamma = 0$ in DDIM (see Appx. G.4.2). In our CFD, the definition of $\gamma$ requires that $\gamma < 1$ (Appx. Eq. 37), which implies a difference between CFD and SDS.

Theoretically, when restricted to 2D image generation where $x = g_\theta(c)$, SDS is equivalent to seeking the maximum likelihood point in the noisy distribution $p_t$ with a Gaussian distribution $\mathcal{N}(\alpha_t x, \sigma_t^2 I)$ centered at the image $x$. When the optimization of SDS loss is near optimal, their generation results are centered around a few modes (Poole et al.,

| | loss gradient | noising method |
|---|---|---|
| SDS | $\epsilon_\phi(x_t) - \epsilon$ | $\epsilon \sim \mathcal{N}(0, I)$ |
| VSD | $\epsilon_\phi(x_t) - \epsilon_{\text{lora}}(x_t)$ | $\epsilon \sim \mathcal{N}(0, I)$ |
| ISM | $\epsilon_\phi(x_t) - \epsilon_\phi(x_s)$ | DDIM inversion($g_\theta(c)$) |
| CFD (ours) | $\epsilon_\phi(x_t) - \tilde{\epsilon}$ | $\tilde{\epsilon} = \tilde{\epsilon}_t(\theta, c)$ |

Table 1: Comparison between score distillation gradients.

2023). In contrast, our CFD is sampling from the whole distribution $p_0$ and equivalent to a diffusion ODE or SDE sampling process with first-order discretization. Thus, CFD can generate more diverse results with better quality.

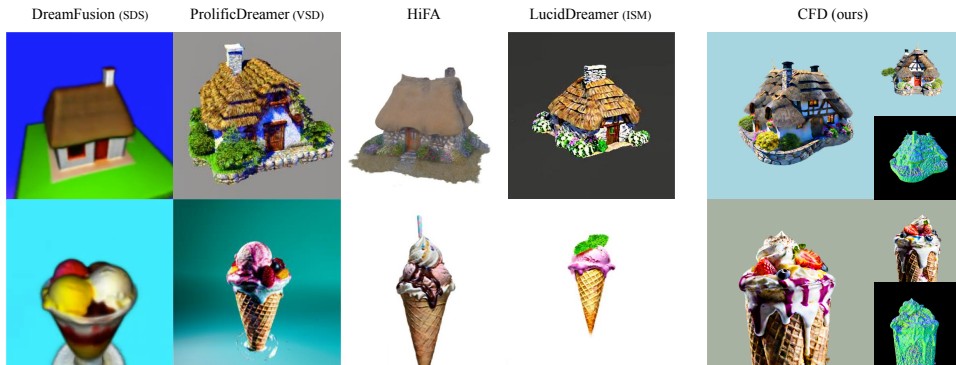

Figure 4: **Visual comparison to baseline methods.** We compare rendered images of our method with baselines include DreamFusion (Poole et al., 2023), ProlificDreamer (Wang et al., 2024a), HiFA (Zhu et al., 2024), LucidDreamer (Liang et al., 2023). The images of baselines are from their official implementations. Prompts: "A 3D model of an adorable cottage with a thatched roof" (top) and "A DSLR photo of an ice cream sundae" (bottom).

**Comparison with other score distillation methods.** We list the loss and noising of different methods in Tab. 1. ISM (Liang et al., 2023) incorporates DDIM inversion noising in their score distillation. While this approach can yield finer details than SDS, computing the inversion significantly increases computational costs. We discuss the connection between our method and ISM in Appx. H. We also list the difference between proposed pipeline and different baseline methods in Appx. E.2.

## 4 EXPERIMENTS

In comparisons to prior methods, we distill Stable Diffusion (Rombach et al., 2022) and use the same codebase threestudio (Guo et al., 2023). We compare CFD with various prior state-of-the-art methods, including SDS (Poole et al., 2023; Wang et al., 2023a), VSD (Wang et al., 2024a) and ISM (Liang et al., 2023). Specifically, VSD incorporates LoRA network training in their score distillation, ISM incorporates DDIM inversion in their score distillation. Since timestep annealing (Zhu et al., 2024; Wang et al., 2024a; Huang et al., 2024) has been shown to help improve generation quality (Wang et al., 2024a; Zhu et al., 2024; Huang et al., 2024), we also apply timestep annealing to all baseline methods. We use results from the official implementation of other baselines in qualitative comparisons if not specified. In addition, we show results of a 2-stage pipeline in Fig. 1(a), 1(b), where we first distill MVDream (Shi et al., 2024), then distill Stable Diffusion, which alleviates the multi-face issue (Poole et al., 2023; Armandpour et al., 2023; Hong et al., 2023a). We provide implementation details in Appx. A and details of experiment metrics in Appx. B.

### 4.1 COMPARISON WITH BASELINES

We compute 3D-FID following VSD (Wang et al., 2024a) to evaluate the quality and diversity of different score distillation methods, and compute 3D-CLIP to evaluate prompt alignment for different methods. We provide qualitative comparison in Fig. 4 and quantitative results in Tab. 2, 3, and Appx. Tab. 5. We also provide additional comparisons with VSD in Appx. Fig. 9, ISM in Appx. Fig. 10, and SDS in Appx. Fig. 11. As shown in both quantitative and qualitative results, CFD outperforms all baseline methods and has better generation quality (Fig. 4 and Appx. Fig. 9, 10, 11) and diversity (Appx. Fig. 9, 10, 11). Our method produces rich details and the results are more photorealistic. Addition results and comparisons are in Appx. C.

|  | 3D-FID $\downarrow$ | 3D-CLIP $\uparrow$ |
|---|---|---|
| SDS | 88.06 | 35.07±0.20 |
| ISM | 86.00 | 34.99±0.26 |
| VSD | 83.02 | 35.10±0.20 |
| CFD (ours) | **78.13** | **35.16±0.23** |

Table 2: **Comparison with baselines on quality, diversity and prompt alignment.** We report averaged clip score of different verison of CLIP backbones. We use 10 seeds for each of the 10 different prompts, respectively.

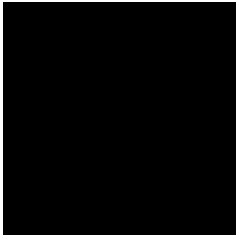 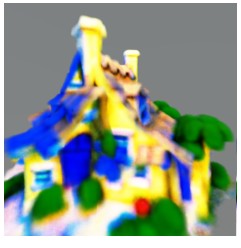 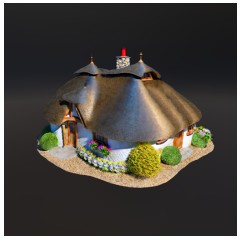 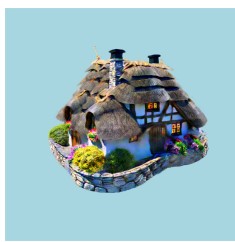

(a) Original PF-ODE      (b) w/ bilinear noise      (c) w/ random noise      (d) w/ consistent noise

Figure 5: **Ablation on the noise design and the flow space.** (a) Directly training $\theta$ with original PF-ODE using Eq. 6 with noisy variable. (b) Distilling with bilinear-interpolated noise map. (c) Distilling with random noise. (d) Distilling with our multi-view consistent Gaussian noise, which has the best visual quality.

| Ranker | Aesthetics | PickScore |
|---|---|---|
| Ours vs. SDS | 0.54 | 0.64 |
| Ours vs. VSD | 0.60 | 0.68 |
| Ours vs. ISM | 0.56 | 0.66 |
| Ours vs. FSD | 0.54 | 0.78 |

Table 3: **Automated win rates comparison under reward models.** We compare the performance of our CFD method against baseline models using Aesthetics Scores (Schuhmann, 2022) and PickScores (Kirstain et al., 2023). Our method consistently achieves a winning rate higher than 0.5, which demonstrates its effectiveness.

## 4.2 ABLATION STUDIES

**Ablation on the flow space.** As shown in Fig. 5: (a) When directly training $\theta$ with original PF-ODE using Eq. 6 with noisy variable, the training fails after several iterations. (b) Simply using bilinear interpolation instead of Noise Transport Equation leads to correlated pixel noise and generates blurry results. (c) When using the random noise as in SDS, the results are over-smoothed. (d) Our consistent flow distillation with multi-view consistent Gaussian noise generates high-quality results. By using a multi-view consistent Gaussian noise, the flow for a fixed camera is more aligned with a diffusion sampling process, and the quality improves. We also provide additional ablations on our design choices in Appx. E.

**Ablation on noise injection rate $\gamma$.** Noise injection rate $\gamma$ in Eq. 11 determines the rate at which new noise will be injected into the noise function. When $\gamma = 0$, no noise will be injected, $\tilde{\epsilon}$ will be fixed constant if the geometry and camera view is fixed and CFD corresponds to using ODE guidance. When $\gamma > 0$, new noise will be injected, and $\tilde{\epsilon}(\theta, c)$ will gradually change. In this case, CFD corresponds to using SDE guidance. Using SDE-based stochastic samplers may help to improve image generation quality as reported in prior works (Song et al., 2021b;a; Karras et al., 2022). In Tab. 4. We also observe that use a small nonzero $\gamma$ helps to improve the performance of CFD. In practice, we found that using a $\gamma$ larger than 0.0001 could result in over-smoothed texture, therefore we set $\gamma = 0.0001$ by default in our experiments for CFD. As a reference, we calculated a typical equivalent $\gamma$ value of DDPM to be $\gamma \approx 0.00024$ (see Appx. G.4.2).

## 5 RELATED WORK

**Diffusion models** Diffusion models (Sohl-Dickstein et al., 2015; Sharma et al., 2018; Ho et al., 2020; Song et al., 2021b; Changpinyo et al., 2021; Schuhmann et al., 2022) are generative models that are learned to reverse a diffusion process. A diffusion process gradually adds noise to a data distribution, and the diffusion model is trained to reverse such an iterative process based on the score function. Denoise Diffusion Implicit Models (DDIM) (Song et al., 2021a) proposed a deterministic

| $\gamma$ | 0.0 | 0.0001 | 0.001 | 0.01 | 1.0 |
|---|---|---|---|---|---|
| 3D-IS ($\uparrow$) | 2.24±0.12 | **2.60±0.21** | 2.47±0.39 | 2.08±0.04 | 1.77±0.13 |

Table 4: **Ablation on noise injection rate** $\gamma$. We ablate the impact of $\gamma$ on 3D generation diversity and quality. We generate samples with 16 random seeds.

tic sampling method to speed up the sampling. Meanwhile, it is proved that a diffusion process corresponds to a Probability Flow Ordinary Differential Equation (PF-ODE) (Song et al., 2021b), which yields the same marginal distributions as the forward diffusion process at any timestep. Later works (Salimans & Ho, 2022; Karras et al., 2022; Lu et al., 2022) demonstrate that DDIM can be viewed as the first-order discretization of the PF-ODE.

**Score distillation sampling** The score distillation sampling (SDS) paradigm for distilling 2D text-to-image diffusion models for 3D generation is proposed in DreamFusion (Poole et al., 2023) and SJC (Wang et al., 2023a). During the distillation process, the learnable 3D representation with differentiable rendering is optimized by the gradient to make the rendered view match the given text. Many recent works follow the SDS paradigm and studied for various aspects, including timestep annealing (Huang et al., 2024; Wang et al., 2024a; Zhu et al., 2024), coarse-to-fine training (Lin et al., 2023; Wang et al., 2024a; Chen et al., 2023), analyzing the components (Katzir et al., 2024), formulation refinement (Zhu et al., 2024; Wang et al., 2024a; Liang et al., 2023; Tang et al., 2023; Wang et al., 2023b; Yu et al., 2024; Armandpour et al., 2023; Wu et al., 2024b; Yan et al., 2024), geometry-texture disentanglement (Chen et al., 2023; Ma et al., 2023; Wang et al., 2024a), addressing multi-face Janus problem replacing the text-to-image diffusion with novel view synthesis diffusion (Liu et al., 2023; Long et al., 2023; Liu et al., 2024b; Weng et al., 2023; Ye et al., 2023; Wang & Shi, 2023) or multi-view diffusion (Shi et al., 2024).

**Reconstruction models** Another prevailing paradigm for 3D generation is to reconstruct the 3D shape given an input image. A typical pipeline is to first generate sparse-view images and then reconstruct the 3D shapes using reconstruction methods (Wu et al., 2024a; Li et al., 2024) or models (Hong et al., 2023b; Liu et al., 2024a; Wang et al., 2024b; Tang et al., 2024). By directly training on relatively large scale 3D dataset like Objaverse (Deitke et al., 2023), these methods are usually capable of generating plausible shapes with a fast speed, but the performance of these models are usually limited when facing out of domain input images.

## 6 CONCLUSION

In this paper, we proposed Consistent Flow Distillation. We begin by leveraging the gradient of the diffusion ODE or SDE sampling process to guide the 3D generation. From a sampling perspective, we identified that using consistent flow to guide the 3D generation is the key to this process. We developed a multi-view consistent Gaussian noise with correct correspondence on the object surface and used it to implement the consistent flow. Our method can generate high-quality 3D representations by distilling 2D image diffusion models and shows improvement in quality and diversity compared with prior score distillation methods.

**Limitations and broader impact.** Although CFD can generate 3D assets of high fidelity and diversity, similar to prior works SDS, ISM, and VSD, the generation can take one to a few hours, and when distilling a text-to-image diffusion model, due to the properties of the teacher models, the distilled 3D representation sometimes may have multi-face Janus problem and may not be good for complex prompt. Besides, due to 3D representation flexibility and interference from other views, it is very hard to guarantee that the sampling process from a rendered view of the 3D object is exactly the same as sampling for 2D images given text in practice. While our 3D consistent noise can reduce the interference and achieve better results, the flow for 3D rendered views may not be exactly the same as 2D flows of the initial noise. Also, like other generative models, it needs to pay attention to avoid generating fake and malicious content.

**Acknowledgements** This work was supported, in part, by the Amazon Research Award, the Qualcomm Innovation Fellowship.

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

# APPENDIX

## A  IMPLEMENTATION DETAILS

In this paper, we conduct experiments primarily on a single NVIDIA-GeForce-RTX-3090 or NVIDIA-L40 GPU. In the quantitative experiments, we adopt similar pipelines (including the choice of 3D representation, training steps, shape initialization, teacher diffusion model, etc.) across methods. We apply timestep annealing for all methods and use the same negative prompts in the quantitative experiments. The main differences between methods lie in the loss functions used.

We use CFG (Ho & Salimans, 2022) scale of 75 for CFD in quantitative experiments. In practice, We found CFD works the best with CFG scale of 50-75. We apply the same fixed negative prompts (Shi et al., 2024; Katzir et al., 2024; McAllister et al., 2024) for different text prompts.

For simple prompts, we directly use CFD to distill Stable Diffusion v2.1 (Fig. 4, 12 and 13).

For mesh generation, we first use CFD to generate coarse shapes with MVDream (Shi et al., 2024). Then we use CFD and follow the geometry and mesh refinement stages in VSD (Wang et al., 2024a) with Stable Diffusion v2.1 to generate the mesh results in Fig. 1(b).

For complex prompts, we adopt a 2 stage pipeline (Fig. 1(a), 1(c), 6, 7, 8, 9, 10 and 11). We first generate coarse shape by distilling MVDream to avoid multi-face problems. Then we distill Stable Diffusion v2.1 to refine the details and colors (stage 2). We randomly replace the rendered image with normal map with 0.2 probability to regularize the geometry in stage 2. The total training time is approximately 3 hours on A100 GPU.

## B  EXPERIMENT DETAILS

**3D-FID**  We compute the FID score between the rendered images for the generated 3D samples and the images generated by the teacher diffusion models following the evaluation setting of VSD (Wang et al., 2024a). For the experiments with 10 prompts in Tab. 2, we sampled 5,000 images for each prompt from Stable Diffusion, creating a real image set with a total of 50,000 images. We generated 3D objects using different score distillation methods, with 10 different seeds per prompt for each method. We rendered 60 views for each 3D object, resulting in a fake image set of 6,000 images. We use FID implementation from torchmetrics package with feature=2048.

**3D-IS**  We compute the Inception Score (IS) for the front-view images to measure the quality and diversity. We set split=2 to compute the standard variance of the IS metric. Due to limited compute budget, we use 16 random seeds for each parameter setting of $\gamma$ and then use the rendered front view to compute IS metric. The IS implementation used in our experiments is from the torchmetrics package.

**3D-CLIP**  We compute the CLIP cosine similarity between the rendered images of the 3D samples and the corresponding text prompt. For one sample, we render 120 views and take the maximum CLIP score. Then we average the CLIP score across different seeds and prompts (and CLIP models). We use CLIP socre implementation from torchmetrics package.

**Aesthetic evaluation**  Following Diffusion-DPO (Wallace et al., 2024), we conduct an automated win rate comparison under reward models in Tab. 3. The performance of our CFD method is evaluated against baseline models using Aesthetics Scores (Schuhmann, 2022) and PickScores (Kirstain et al., 2023). We calculate the scores on rendered images generated from 50 samples, each corresponding to a randomly selected prompt.

## C  ADDITIONAL QUALITATIVE COMPARISON

We present more comparison between baseline methods and CFD in Fig. 9, Fig. 10, and Fig. 11. We present additional generation results of CFD in Fig. 6, Fig. 7, Fig. 8, and Fig. 12.

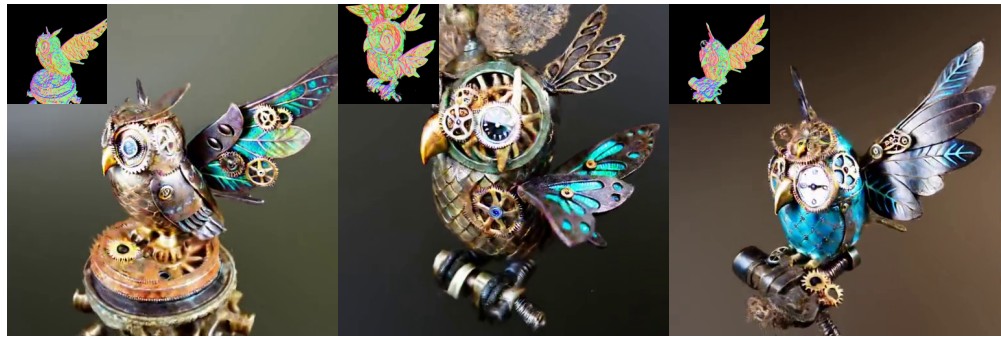

"A steampunk owl with mechanical wings"

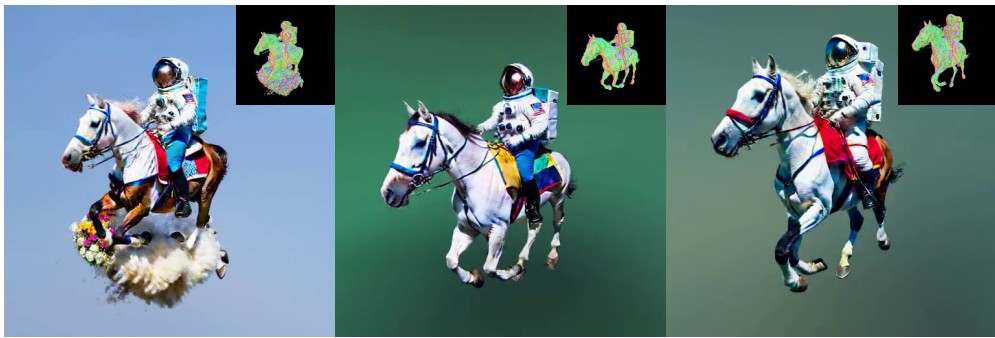

"An astronaut riding a horse"

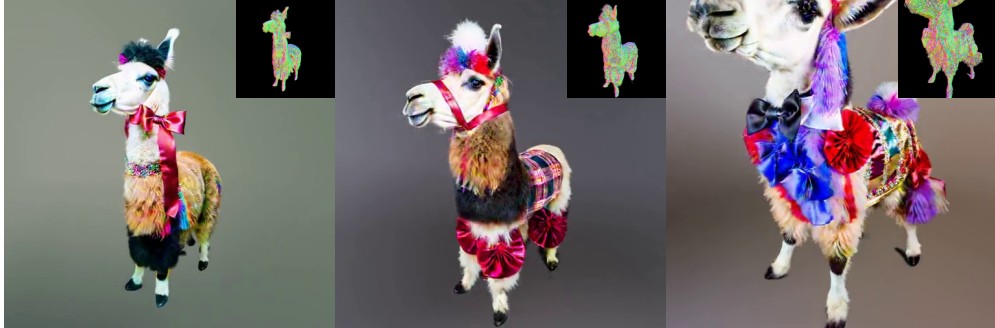

"A llama in a tuxedo at a fancy gala"

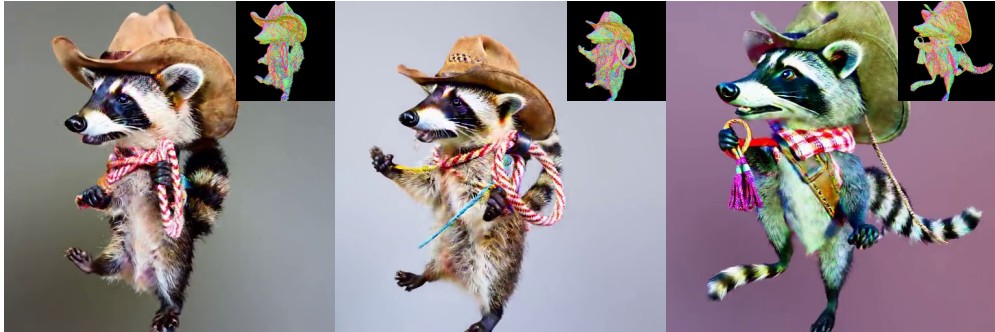

"A cowboy raccoon with a lasso"

Figure 6: **Diverse NeRF results of CFD distilling MVDream then Stable Diffusion (Rombach et al., 2022) on complex prompts.**

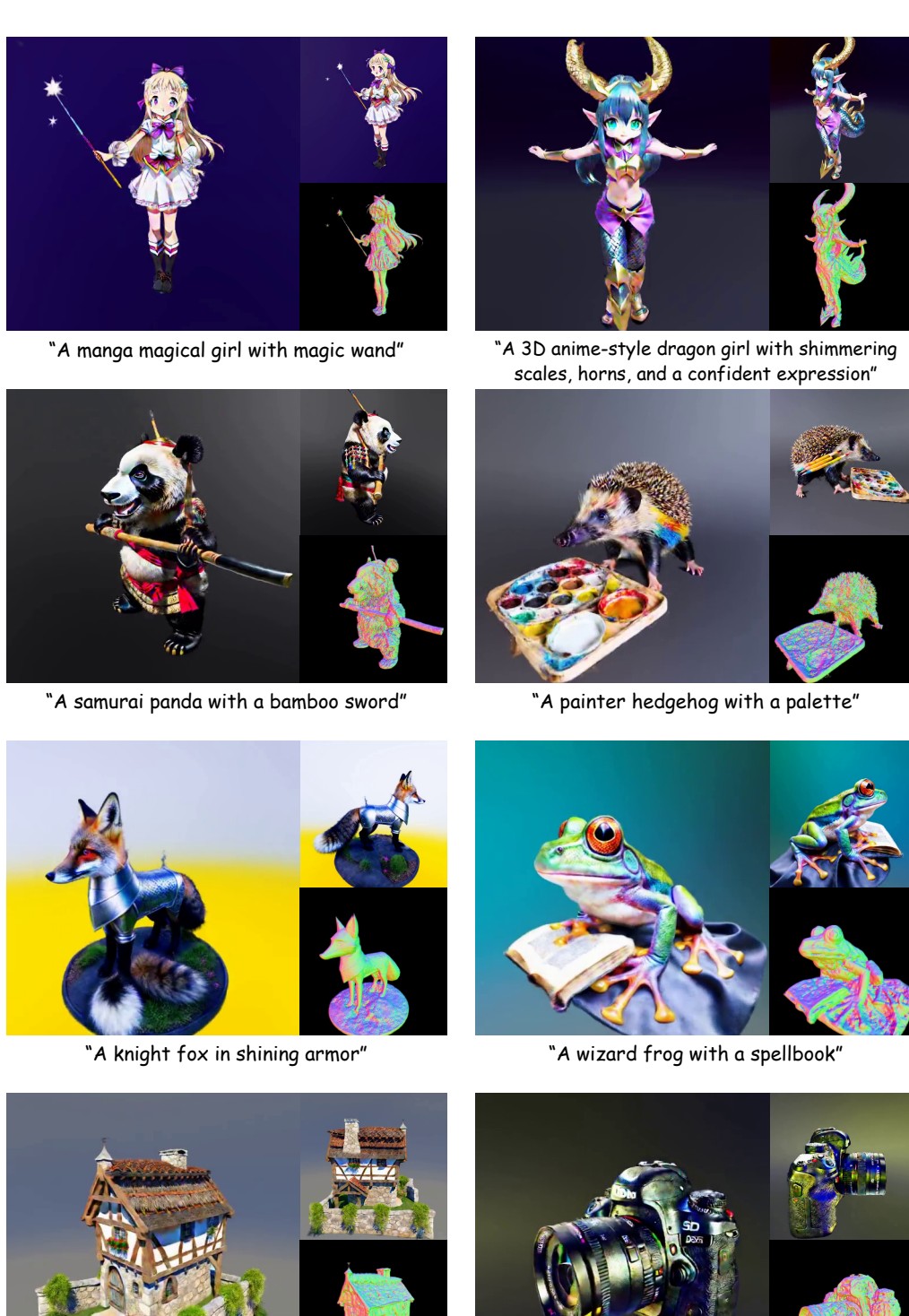

"A manga magical girl with magic wand"

"A 3D anime-style dragon girl with shimmering scales, horns, and a confident expression"

"A samurai panda with a bamboo sword"

"A painter hedgehog with a palette"

"A knight fox in shining armor"

"A wizard frog with a spellbook"

"A 3D model of a medieval house with grass, vines, stone, wood, and medieval decor"

"A 3D model of a DSLR camera, photography, box modeling, Maya"

Figure 7: **NeRF results of CFD distilling MVDream then Stable Diffusion (Rombach et al., 2022) on complex prompts.**

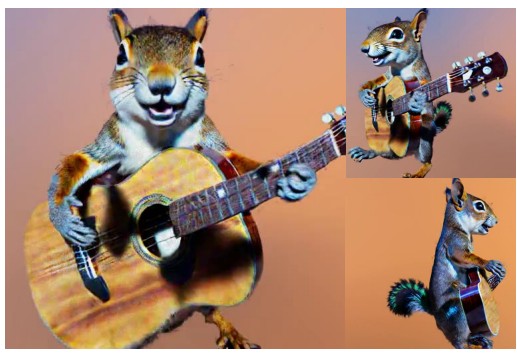

"A squirrel playing the guitar"

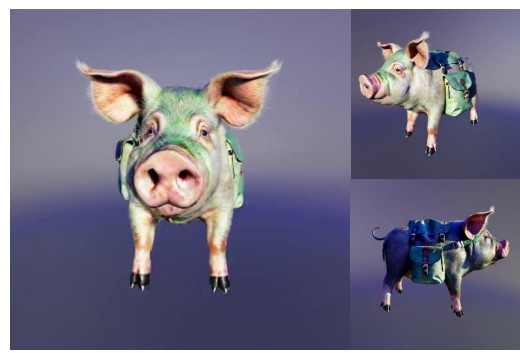

"A pig wearing a backpack"

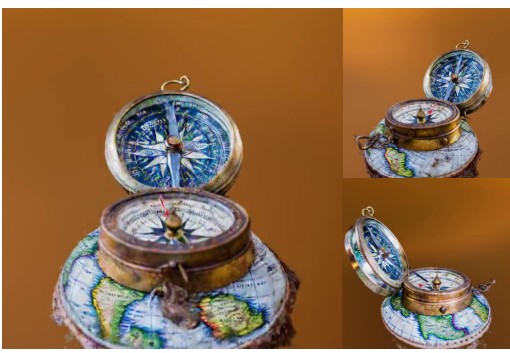

"A weathered brass compass with a cracked glass face, resting
on an old, map. The compass is slightly tarnished, showing signs
of age, bathed in soft, diffused sunlight."

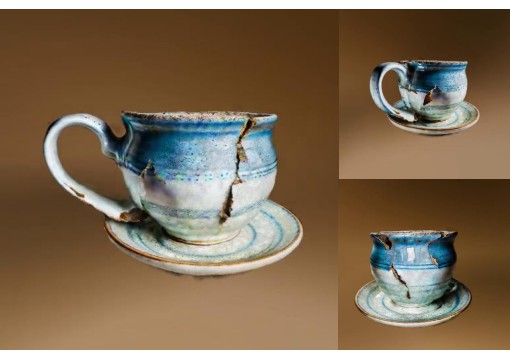

"A cracked ceramic mug, chipped along the rim and faded from
years of use, resting on a rustic wooden table, with morning
sunlight casting soft shadows across its surface."

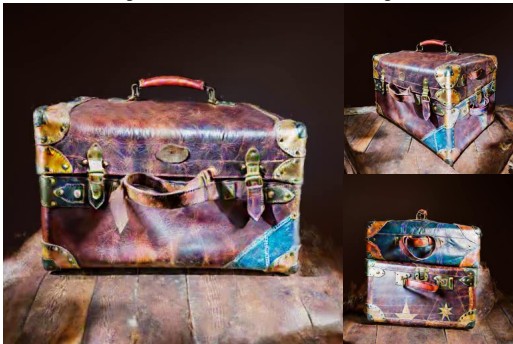

"An old leather suitcase, its corners frayed and its surface
marked with age, labeled with vintage travel tags, placed on a
wooden floor bathed in soft light."

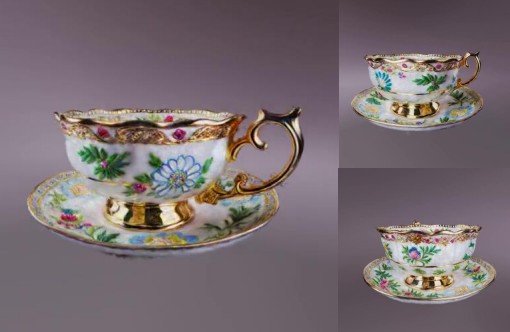

"A delicate porcelain teacup with a gold-rimmed edge, resting on an
embroidered tablecloth. Soft light gleams off the fine china, revealing
its intricate floral design and subtle cracks from age."

Figure 8: **NeRF results of CFD distilling MVDream then Stable Diffusion (Rombach et al.,
2022) on complex prompts.** CFD successfully generated multiple objects and most align with long
prompts.

## D    ALGORITHMS

We provide pseudo algorithms for CFD in Algorithm 1. Algorithm 2 presents how to compute the
multi-view consistent Gaussian noise $\tilde{\epsilon}(\theta, \boldsymbol{c})$.

**Choices of warping function $\mathcal{T}^{-1}$ and reference space $E_{ref}$**    Generally speaking, correct corre-
spondence of noise map between different camera views can be achieved with any choice of con-
tinuous warping function $\mathcal{T}^{-1}$ and reference space $E_{ref}$. In this work, we choose $E_{ref}$ to be a 2D
square space $E_{ref} = [-1, 1]^2$ to utilize existing fast rasterization algorithms, so that Algorithm 2
can be efficiently computed. We design a warping function $\mathcal{T}^{-1}$ to map points in 3D world space
$E_{world}$ to 2D reference space $E_{ref}$. Specifically, to compute the warping $\mathcal{T}^{-1}$ we first convert the

VSD                    CFD (ours)

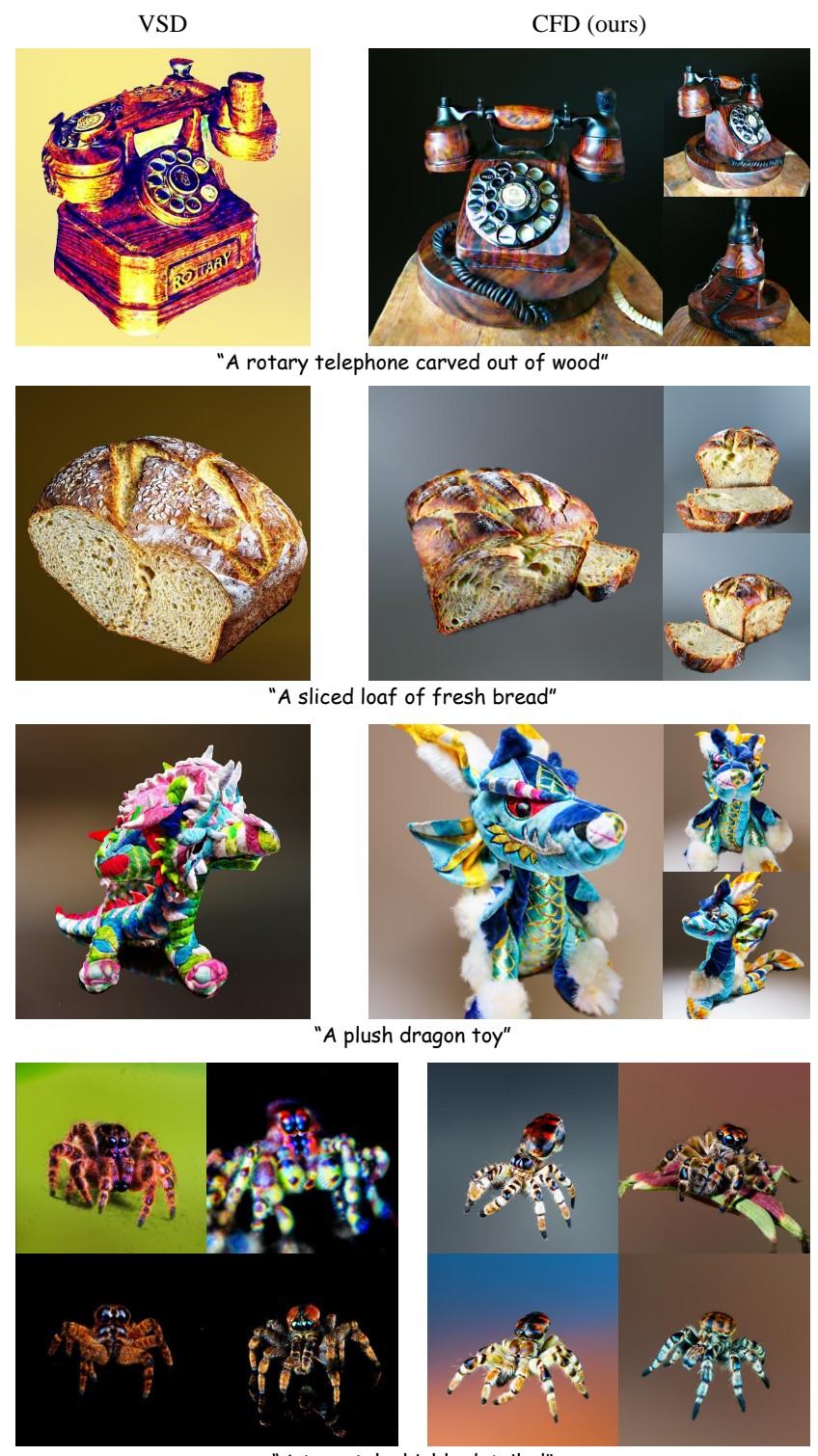

"A rotary telephone carved out of wood"

"A sliced loaf of fresh bread"

"A plush dragon toy"

"A tarantula, highly detailed"

Figure 9: **Additional comparison with ProlificDreamer (VSD) (Wang et al., 2024a).** We use results from the official implementation of the baseline. We show generation results of different methods with different seeds in the last row.

ISM                                    CFD (ours)

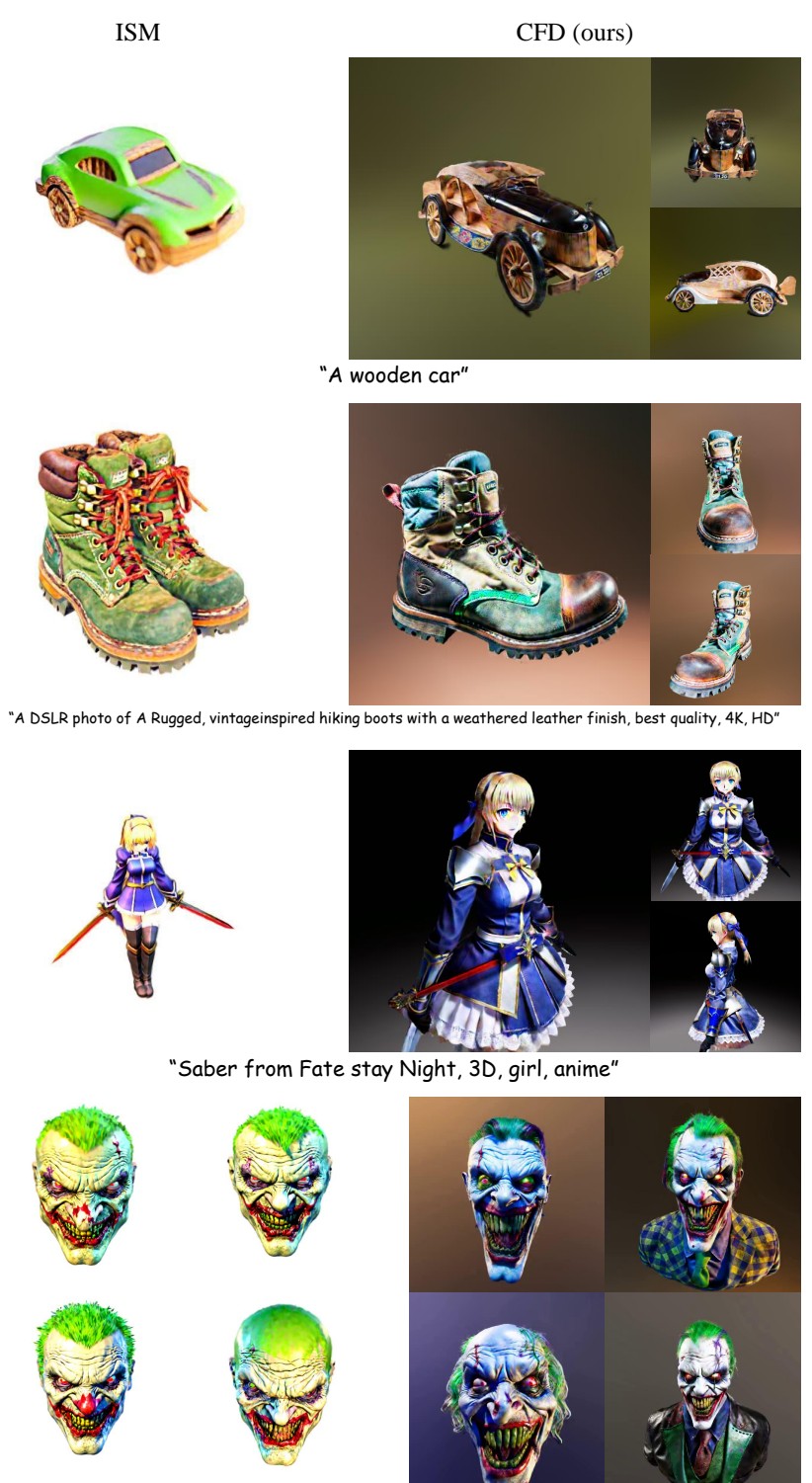

"A wooden car"

"A DSLR photo of A Rugged, vintageinspired hiking boots with a weathered leather finish, best quality, 4K, HD"

"Saber from Fate stay Night, 3D, girl, anime"

"Zombie JOKER, head, HDR, photorealistic, 8K"

Figure 10: **Additional comparison with LucidDreamer (ISM) (Liang et al., 2023).** We use results from the official implementation of the baseline. We show generation results of different methods with different seeds in the last row.

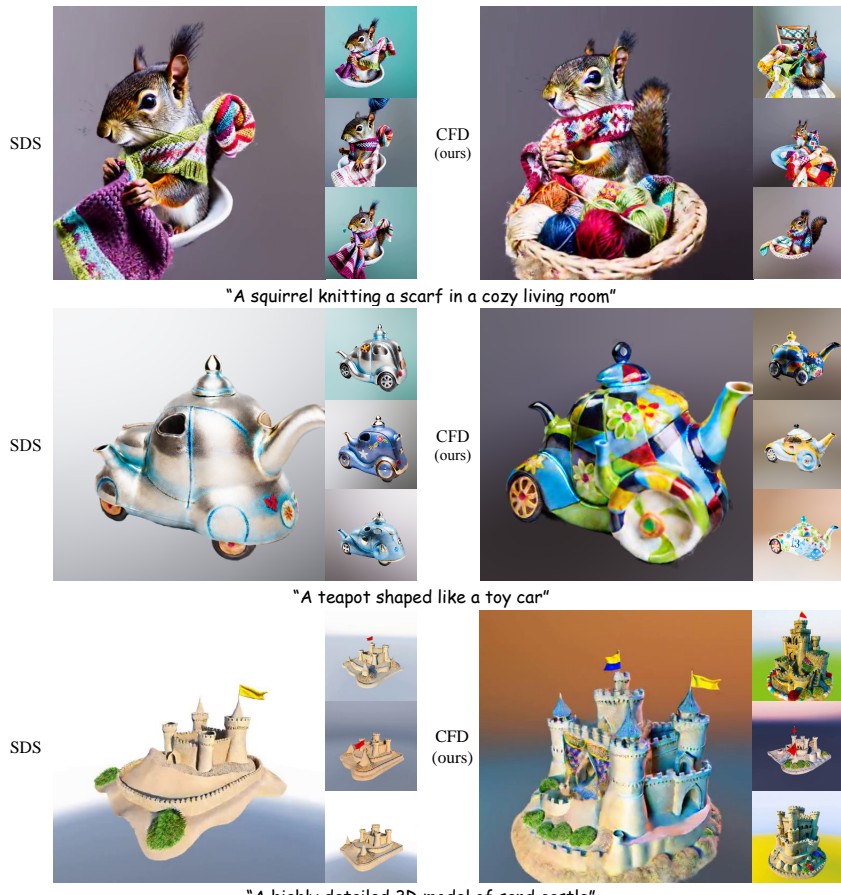

Figure 11: **Comparison with SDS.** We distill MVDream (Shi et al., 2024) and Stable Diffusion (Rombach et al., 2022) in this experiment. We first generate coarse shape by distilling MVDream using SDS and CFD, then distill Stable Diffusion to refined the color with SDS and CFD, respectively. In this figure, the only difference between two methods is the noise function used by SDS and CFD. We use 4 different seeds for each methods in this figure. SDS trends to generate oversmoothed textures and identical simple shapes. CFD outperforms SDS with better diversity and fidelity.

|  | 3D-CLIP ↑ | | | |
|---|---|---|---|---|
|  | B16 | B32 | L14 | L14-336 |
| SDS | 36.30 | 35.99 | 31.82 | 32.42 |
| VSD | 36.58 | 36.27 | 31.97 | 32.67 |
| CFD (ours) | **36.79** | **36.32** | **32.44** | **33.10** |

Table 5: **Comparison with baselines on prompt alignment.** We use 1 random seed for each of the 128 prompts. B16, B32, L14, L14-336 denote different versions of CLIP backbones. We observe that CFD is competitive or outperform SDS and VSD on prompt alignment.

points at $(x_p, y_p, z_p)$ to spherical coordinates $(r_p, \theta_p, \phi_p)$. For simplicity, we only present the case when $\phi_p \in [0, \frac{\pi}{2})$. The point is then mapped to $(x_r, y_r) \in E_{ref}$, where

$$\begin{cases} x_r = \sqrt{1 - \cos\theta_p}, \\ y_r = \sqrt{1 - \cos\theta_p} \cdot (2 \cdot \frac{\phi_p}{\frac{\pi}{2}} - 1). \end{cases} \quad (14)$$

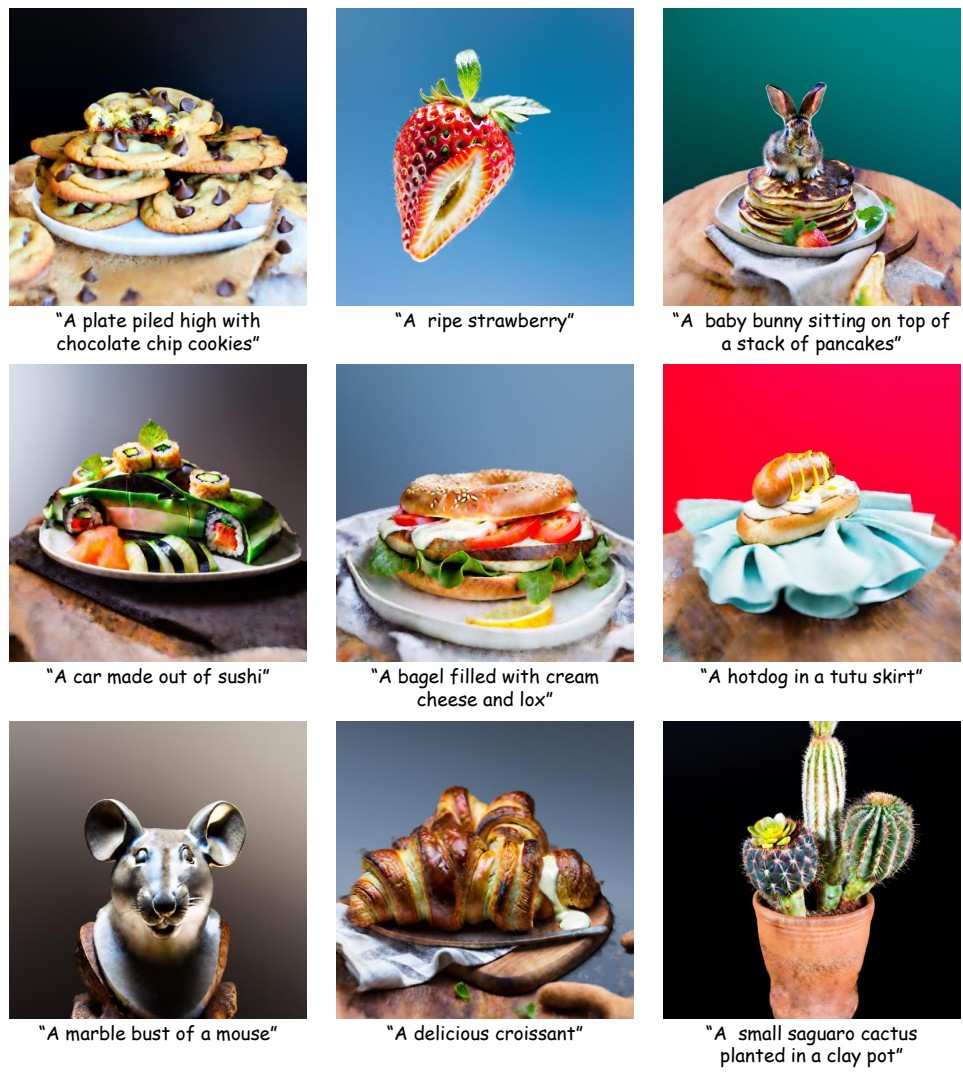

Figure 12: **NeRF results of CFD distilling Stable Diffusion (Rombach et al., 2022).**

Under this mapping function, one can verify that $\mathrm{d}x_r\mathrm{d}y_r = |\frac{\partial(x_r,y_r)}{\partial(\theta_p,\phi_p)}|\mathrm{d}\theta_p\mathrm{d}\phi_p = \frac{2}{\pi}\cdot\sin\theta_p\mathrm{d}\theta_p\mathrm{d}\phi_p$. So points uniformly scattered on the sphere in 3D space $E_{world}$ will remain uniform after being mapped to the reference 2D space $E_{ref}$. This design helps to improve the fairness of Algorithm 2 so that we can use a lower resolution reference space while keeping most of the warped triangles covering enough area in the reference space $E_{ref}$. Notably, two different triangles could overlap with the warping defined by Eq. 14, resulting in correlations across the pixels of the computed noise function $\tilde{\epsilon}(\theta, \boldsymbol{c})$ in the same camera view. This overlap occurs only when the surface of the 3D object intersects the radius of a sphere centered at the origin of the $E_{world}$ more than once. However, we do not observe the destructive effects seen in other interpolation methods that can lead to correlation between pixels (as in Fig. 5 (b)) in our experiments, and we believe it is unnecessary to find a warping function that avoids such overlapping completely.

**Reference space $E_{ref}$ resolution**   We use reference space with resolution of $2048 \times 2048$ in most of our experiments. This will only introduce $8.1\%$ computation overhead to our training (tested on RTX-3090 GPU). The teacher model Stable Diffusion represents the whole object with latent at 64 resolution and MVDream (Shi et al., 2024) 32, so noise map with 2048 resolution is sufficient. We also observe the quality is similar with noise map resolutions from 512 to 2048.

| SDS | VSD | FSD | CFD (ours) |
|---|---|---|---|

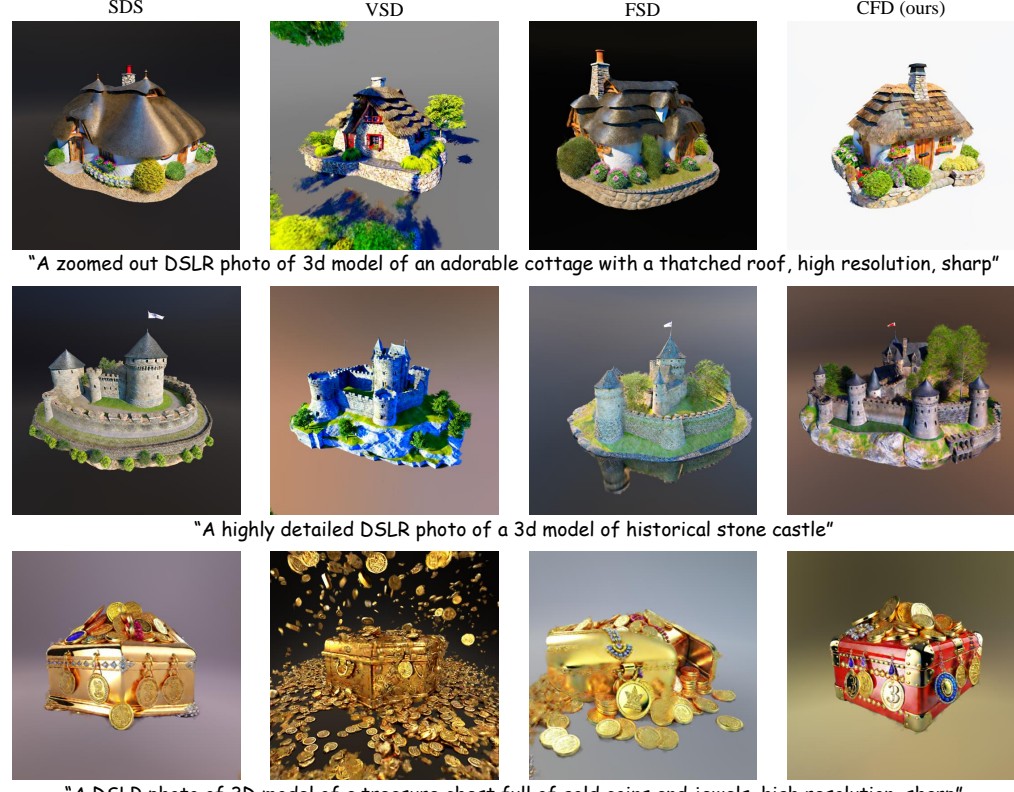

"A zoomed out DSLR photo of 3d model of an adorable cottage with a thatched roof, high resolution, sharp"

"A highly detailed DSLR photo of a 3d model of historical stone castle"

"A DSLR photo of 3D model of a treasure chest full of gold coins and jewels, high resolution, sharp"

Figure 13: **Comparison with SDS, VSD and FSD.** We distill Stable Diffusion (Rombach et al., 2022) with different score distillation methods in this experiment. CFD outperforms SDS, VSD and FSD with better visual quality, geometry and has richer details.

# E    ADDITIONAL ABLATIONS

## E.1    ABLATION ON THE DESIGN SPACE

We ablate our proposed improvement step by step in this section. Timestep annealing (Wang et al., 2024a; Zhu et al., 2024; Huang et al., 2024) is helpful for forming finer details. Adding negative prompts (Shi et al., 2024; Katzir et al., 2024; McAllister et al., 2024) helps to improve generation styles. We also find that adding negative prompts is crucial when timestep $t(\tau)$ is small. Without negative prompts, the color of samples will become unnatural during the optimization at small timesteps. In this work, we apply negative prompts by directly replacing the unconditional prediction of the diffusion model with prediction conditioned on negative prompts. Finally, by changing the random sampled noise in SDS with our multi-view consistent Gaussian noise, the generated samples can form much richer details and are more diverse. We visualize this ablation in Fig. 15.

We propose utilizing CFD to distill the multi-view diffusion model, MVDream, in Stage 1 as shape initialization for complex prompts. This decision is based on our observation that both baseline methods and our CFD can experience multi-face issues when solely distilling SDv2.1 (Fig. 14(a) and Fig. 14(b)). However, distilling only MVDream produces low-quality results (Fig. 14(c)). To address these issues, we adopt a two-stage pipeline in our complete method, where Stage 1 initializes the shape using MVDream, and Stage 2 refines it by distilling SDv2.1. This approach effectively mitigates the challenges identified above, as illustrated in Fig. 14(d).

## E.2    COMPARE THE PIPELINE OF DIFFERENT METHODS

We list the differences between the pipelines of different baseline methods in Tab. 7.

---

**Algorithm 1** CFD

---

1: **Input:** 3D representation parameter $\theta$, prompt $y$, pretrained diffusion model $\epsilon_\phi(\boldsymbol{x}_t, t, y)$, render $\boldsymbol{g}_\theta(\boldsymbol{c})$, annealing time-schedule $t(\tau)$, learning rate $lr$.
2: **Output:** 3D representation parameter $\theta$.
3: **for** $\tau$ from 0 to $\tau_{end}$ **do**
4:     Sample camera view $\boldsymbol{c}$
5:     Render image $\boldsymbol{g}_\theta(\boldsymbol{c})$, depth map $\boldsymbol{Depth}(\boldsymbol{c})$, and opacity map $\boldsymbol{Opacity}(\boldsymbol{c})$
6:     Get diffusion timestep $t(\tau)$
7:     Compute 3D Consistent Noise $\tilde{\epsilon}(\theta, \boldsymbol{c})$                  ▷ Refer to Algorithm 2
8:     $\boldsymbol{x}_t \leftarrow \alpha_t \boldsymbol{g}_\theta(\boldsymbol{c}) + \sigma_t \tilde{\epsilon}(\theta, \boldsymbol{c})$
9:     $\theta \leftarrow \theta - lr \cdot (\epsilon_\phi(\boldsymbol{x}_t, t(\tau), y) - \tilde{\epsilon}(\theta, \boldsymbol{c})) \frac{\partial \boldsymbol{g}_\theta(\boldsymbol{c})}{\partial \theta}$
10: **end for**

---

**Algorithm 2** Computing 3D Consistent Noise

---

1: **Initialization:** Noise background $\epsilon_{bg}$, high resolution noise $\epsilon_{ref}$ in reference space $E_{ref}$, opacity threshold $o_{th}$, noise injection rate $\gamma$.
2: **Input:** Depth map $\boldsymbol{Depth}(\boldsymbol{c})$, opacity map $\boldsymbol{Opacity}(\boldsymbol{c})$.
3: **Output:** $\tilde{\epsilon}(\theta, \boldsymbol{c}) = \epsilon_{out}$.
4: Triangulate the pixels to $\boldsymbol{p}$
5: Project those triangles to the surface $\boldsymbol{ctw}(\boldsymbol{p})$ in world space $E_{world}$ according to $\boldsymbol{Depth}(\boldsymbol{c})$
6: Warp the triangles from world space $E_{world}$ to reference space $E_{ref}$ as $\mathcal{T}^{-1}(\boldsymbol{ctw}_{\boldsymbol{c}}(\boldsymbol{p}))$
7: Rasterize and aggregate the noise values on $\epsilon_{ref}$ corvered by the triangles
8: $\epsilon_{out} \leftarrow \epsilon_{bg}$
9: $\epsilon_{out}[\boldsymbol{Opacity}(\boldsymbol{c}) > o_{th}]_{\boldsymbol{p}} \leftarrow \frac{1}{\sqrt{n}} \sum_{(x,y)_i \text{ covered by the rasterized triangle } \mathcal{T}^{-1}(\boldsymbol{ctw}_{\boldsymbol{c}}(\boldsymbol{p}))}^{n} \epsilon_{ref}[x, y]$
10: **if** $\gamma > 0$ **then**
11:     $\epsilon_{bg} \leftarrow \sqrt{1-\gamma}\epsilon_{bg} + \sqrt{\gamma} \cdot \text{randn\_like}(\epsilon_{bg})$           ▷ SDE noise injection
12:     $\epsilon_{ref} \leftarrow \sqrt{1-\gamma}\epsilon_{ref} + \sqrt{\gamma} \cdot \text{randn\_like}(\epsilon_{ref})$
13: **end if**
14: Return $\epsilon_{out}$

---

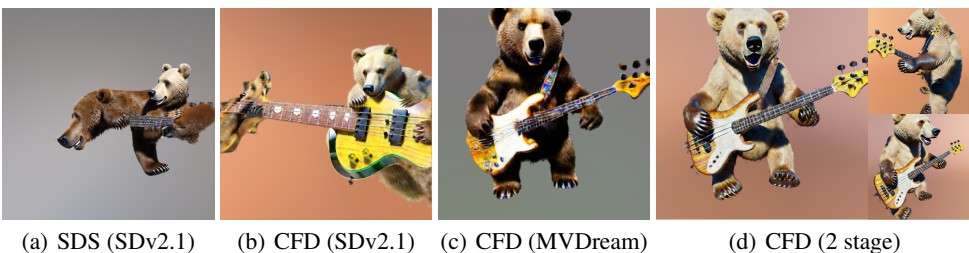

(a) SDS (SDv2.1)     (b) CFD (SDv2.1)     (c) CFD (MVDream)     (d) CFD (2 stage)

Figure 14: **Ablation on the pipeline stages.** Prompt: "A bear playing an electric bass".

### E.3 COMPARISON ON NOISE METHODS

We list the differences between the noising methods of different baseline methods in Tab. 6. Concurrent work FSD (Yan et al., 2024) also employs a deterministic, view-dependent noising function and can therefore be considered a special case of our CFD with $\gamma = 0$. The noise of FSD is aligned on a shpere independent of the 3D object surface. However, this noise design can still lead to over-smoothed textures, and the misalignment of noise with the 3D object surface can sometimes result in suboptimal geometry (see Fig. 13). The noise design of FSD is inferior to ours when the 3D object shape is nearly formed. Gradient consistency is essential for accurately constructing geometry in differentiable 3D representations like NeRF. Aligning noise in 3D space independently of the object surface can lead to deviations from the original geometry, even when a relatively good shape

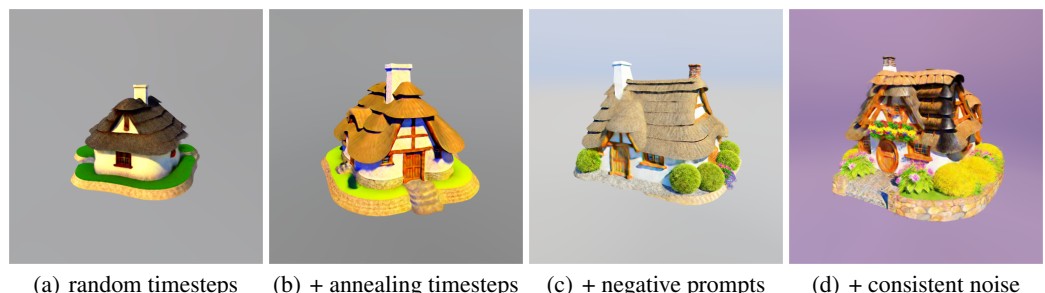

(a) random timesteps     (b) + annealing timesteps     (c) + negative prompts     (d) + consistent noise

Figure 15: **Ablation on the proposed improvements.**

| | VSD | ISM | CFD (ours) |
|---|---|---|---|
| **Timestep schedule** | | (sample $t \sim \mathcal{U}(t_{min}, t_{max})$) | |
| $t_{max} = t_{min}$ | False | False | True |
| $t_{max}$ | Abrupt decrease | Linearly decrease (till $t_0$) | Linearly decrease (multi-stage) |
| $t_{min}$ | Fixed | Fixed | Linearly decrease (multi-stage) |
| **Noise** | | | |
| Noise type | Random | Inversion | Consistent |
| **3D representation** | | | |
| Shape initialization | Stable Diffusion(+VSD) | point-e | MVDream(+CFD) |
| Representation | NeRF→Mesh | point cloud→3DGS | NeRF(→Mesh) |
| **Uncond prompt** | | | |
| LoRA network | True | False | False |
| Negative prompt | False | True | True |

Table 6: Comparison between pipelines of VSD, ISM and CFD.

is formed, as a highly consistent region may be located away from the surface. In contrast, our noise design, which aligns with the object surface, avoids such issues.

Notably, the object surface can slowly change during the generation process, so the noise for the same view in CFD is not strictly fixed even when $\gamma = 0$ in Eq. 11.

## F    GRADIENT VARIANCE

We compare the gradient variance of different methods during training. We compute the scaled gradient variance by taking Exponential Moving Average parameters $\hat{v}_t$, $\hat{m}_t$ from Adam optimizer for convenience. We report the scaled gradient variance $\sigma$ on the parameters of nerf hash encoding with 10 seeds for each of the noising methods. $\sigma$ was calculated according to (where $g_t$ is the gradient):

$$\begin{cases} \hat{m}_t \approx \mathbb{E}[g_t], \\ \hat{v}_t \approx \mathbb{E}[g_t^2], \\ \sigma = \sqrt{\frac{\text{sum}(\hat{v}_t - \hat{m}_t^2)}{\text{sum}(\hat{v}_t)}} \approx \sqrt{\frac{\text{sum}(\text{Var}(g_t))}{\text{sum}(\hat{v}_t)}}. \end{cases} \tag{15}$$

We report the gradient variance in training for VSD (Wang et al., 2024a), SDS (Poole et al., 2023; Wang et al., 2023a), FSD (Yan et al., 2024) and our methods in Tab. 8.

|  | SDS | FSD | CFD (ours, when $\gamma = 0$) |
|---|---|---|---|
| **Timestep schedule** | Random | Annealing | Annealing |
| **Same view noise** | Random | Fixed | Mostly fixed (surface-dependent) |
| **Different views noise** | Independent | Aligned on sphere | Aligned on object surface |

Table 7: Comparision between SDS, FSD, and CFD.

|  | VSD | SDS | FSD | CFD (ours) |
|---|---|---|---|---|
| $\sigma$ ($\downarrow$) | 5.165$\pm$ 0.458 | 4.670$\pm$0.066 | 4.580$\pm$0.081 | **4.521$\pm$0.090** |

Table 8: **Scaled Gradient Variance.** Our CFD has the lowest gradient variance.

# G    CLEAN FLOW SDE

## G.1    BACKGROUND

Song et al. (Song et al., 2021b) presented a SDE that has the same marginal distribution $p_t(\boldsymbol{x}_t)$ as the forward diffusion process (Eq. 1). EDM (Karras et al., 2022) presented a more general form of this SDE, and the SDE corresponds to forward process defined in Eq. 1 takes the following form:

$$\mathrm{d}\big(\frac{\boldsymbol{x}_\pm}{\alpha_t}\big) = -\sigma_t\nabla_{\boldsymbol{x}_\pm}\log p_t(\boldsymbol{x}_\pm)\mathrm{d}\big(\frac{\sigma_t}{\alpha_t}\big) \pm \beta_t\big(\frac{\sigma_t}{\alpha_t}\big)\sigma_t\nabla_{\boldsymbol{x}_\pm}\log p_t(\boldsymbol{x}_\pm)\mathrm{d}t + \sqrt{2\beta_t}\big(\frac{\sigma_t}{\alpha_t}\big)\mathrm{d}\boldsymbol{w}_t \quad (16)$$

$$= \big(\mathrm{d}\big(\frac{\sigma_t}{\alpha_t}\big) \mp \frac{\sigma_t}{\alpha_t}\beta_t\mathrm{d}t\big) \cdot \underbrace{\boldsymbol{\epsilon}_\phi(\boldsymbol{x}_\pm, t, y)}_{-\sigma_t\nabla_{\boldsymbol{x}}\log p_t(\boldsymbol{x})} + \sqrt{2\beta_t}\big(\frac{\sigma_t}{\alpha_t}\big)\mathrm{d}\boldsymbol{w}_t, \quad (17)$$

where $\mathrm{d}\boldsymbol{w}_t$ is the standard Wiener process. If we set $\alpha_t = 1$ for all $t \in [0, T]$, Eq. 17 will become the same SDE in EDM (Karras et al., 2022). The initial condition for the forward process is $\boldsymbol{x}_+ \sim p_{t_s}(\boldsymbol{x}_+)$ at $t = t_s$ ($t_s$ is small enough but $t_s > 0$ to avoid numerical issues), and for the reverse process, it is $\boldsymbol{x}_- \sim \mathcal{N}(\boldsymbol{0}, \sigma_T^2\boldsymbol{I})$ at $t = T$ (Note that we also let $\alpha_T$ be a small number but $\alpha_T > 0$ to avoid numerical issues).

## G.2    CLEAN FLOW SDE

The clean flow SDE takes the following form:

$$\begin{cases} \mathrm{d}\hat{\boldsymbol{x}}_\pm^c = \big(\mathrm{d}\big(\frac{\sigma_t}{\alpha_t}\big) \mp \frac{\sigma_t}{\alpha_t}\beta_t\mathrm{d}t\big) \cdot \big(\boldsymbol{\epsilon}_\phi(\alpha_t\hat{\boldsymbol{x}}_\pm^c + \sigma_t\tilde{\boldsymbol{\epsilon}}_\pm, t, y) - \tilde{\boldsymbol{\epsilon}}_\pm\big), \\ \mathrm{d}\tilde{\boldsymbol{\epsilon}}_\pm = \mp\tilde{\boldsymbol{\epsilon}}_\pm\beta_t\mathrm{d}t + \sqrt{2\beta_t}\mathrm{d}\boldsymbol{w}_t, \end{cases} \quad (18)$$

where $\mathrm{d}\boldsymbol{w}_t$ is the standard Wiener process. For the forward process, the initial condition at $t = t_s$ is $\hat{\boldsymbol{x}}_+^c \sim p_0(\boldsymbol{x}_+)$, $\tilde{\boldsymbol{\epsilon}}_+ \sim \mathcal{N}(\boldsymbol{0}, \boldsymbol{I})$, and $\hat{\boldsymbol{x}}_+^c$ and $\tilde{\boldsymbol{\epsilon}}_+$ are independent. For the reverse process, the initial condition at $t = T$ is $\hat{\boldsymbol{x}}_-^c = \boldsymbol{0}$ and $\tilde{\boldsymbol{\epsilon}}_- \sim \mathcal{N}(\boldsymbol{0}, \boldsymbol{I})$.

**Proposition 1** (Clean flow SDE is equivalent to diffusion SDE). *In Eq. 18, if we define a new variable $\boldsymbol{x}_\pm'$ according to*

$$\boldsymbol{x}_\pm' = \alpha_t\hat{\boldsymbol{x}}_\pm^c + \sigma_t\tilde{\boldsymbol{\epsilon}}_\pm, \quad (19)$$

*then $\boldsymbol{x}_\pm'$ and $\boldsymbol{x}_\pm$ in Eq. 17 have the same law (probability distribution) for all $t \in [t_s, T]$. i.e. Eq. 18 and Eq. 17 are equivalent.*

*proof.* We prove the equivalence by showing that the initial conditions and dynamics for $\boldsymbol{x}_\pm'$ and $\boldsymbol{x}_\pm$ are identical.

**Initial conditions.** For the forward process of Eq. 18 at $t = t_s$, $\boldsymbol{x}_+' = \alpha_{t_s}\hat{\boldsymbol{x}}_+^c + \sigma_{t_s}\tilde{\boldsymbol{\epsilon}}_+$. Thus, $\boldsymbol{x}_\pm' \sim p_{t_s}(\boldsymbol{x}_\pm')$ according to the definition of a forward diffusion process (Eq. 1). For the reverse process of Eq. 18 at $t = T$, $\boldsymbol{x}_-' = \alpha_T \cdot \boldsymbol{0} + \sigma_T\tilde{\boldsymbol{\epsilon}}_- = \sigma_T\tilde{\boldsymbol{\epsilon}}_-$. So $\boldsymbol{x}_-' \sim \mathcal{N}(\boldsymbol{0}, \sigma_T^2\boldsymbol{I})$.

**Dynamics.** The dynamic of $\boldsymbol{x}'_\pm$ can be derived according to:

$$
\begin{aligned}
\mathrm{d}\big(\frac{\boldsymbol{x}'_\pm}{\alpha_t}\big) =& \mathrm{d}\big(\hat{\boldsymbol{x}}^{\mathrm{c}}_\pm + \frac{\sigma_t}{\alpha_t}\tilde{\boldsymbol{\epsilon}}_\pm\big) \\
=& \mathrm{d}\hat{\boldsymbol{x}}^{\mathrm{c}}_\pm + \mathrm{d}\big(\frac{\sigma_t}{\alpha_t}\big)\tilde{\boldsymbol{\epsilon}}_\pm + \frac{\sigma_t}{\alpha_t}\mathrm{d}\tilde{\boldsymbol{\epsilon}}_\pm \\
=& \big(\mathrm{d}\big(\frac{\sigma_t}{\alpha_t}\big) \mp \frac{\sigma_t}{\alpha_t}\beta_t\mathrm{d}t\big) \cdot \big(\boldsymbol{\epsilon}_\phi(\alpha_t\hat{\boldsymbol{x}}^{\mathrm{c}}_\pm + \sigma_t\tilde{\boldsymbol{\epsilon}}_\pm, t, y) - \tilde{\boldsymbol{\epsilon}}_\pm\big) + \mathrm{d}\big(\frac{\sigma_t}{\alpha_t}\big)\tilde{\boldsymbol{\epsilon}}_\pm + \frac{\sigma_t}{\alpha_t}\mathrm{d}\tilde{\boldsymbol{\epsilon}}_\pm \\
=& \big(\mathrm{d}\big(\frac{\sigma_t}{\alpha_t}\big) \mp \frac{\sigma_t}{\alpha_t}\beta_t\mathrm{d}t\big) \cdot \big(\boldsymbol{\epsilon}_\phi(\boldsymbol{x}'_\pm, t, y) - \tilde{\boldsymbol{\epsilon}}_\pm\big) + \mathrm{d}\big(\frac{\sigma_t}{\alpha_t}\big)\tilde{\boldsymbol{\epsilon}}_\pm + \frac{\sigma_t}{\alpha_t}\mathrm{d}\tilde{\boldsymbol{\epsilon}}_\pm \\
=& \big(\mathrm{d}\big(\frac{\sigma_t}{\alpha_t}\big) \mp \frac{\sigma_t}{\alpha_t}\beta_t\mathrm{d}t\big) \cdot \boldsymbol{\epsilon}_\phi(\cdots) - \mathrm{d}\big(\frac{\sigma_t}{\alpha_t}\big)\tilde{\boldsymbol{\epsilon}}_\pm \pm \frac{\sigma_t}{\alpha_t}\beta_t\tilde{\boldsymbol{\epsilon}}_\pm\mathrm{d}t + \mathrm{d}\big(\frac{\sigma_t}{\alpha_t}\big)\tilde{\boldsymbol{\epsilon}}_\pm + \frac{\sigma_t}{\alpha_t}\mathrm{d}\tilde{\boldsymbol{\epsilon}}_\pm \\
=& \big(\mathrm{d}\big(\frac{\sigma_t}{\alpha_t}\big) \mp \frac{\sigma_t}{\alpha_t}\beta_t\mathrm{d}t\big) \cdot \boldsymbol{\epsilon}_\phi(\cdots) \pm \frac{\sigma_t}{\alpha_t}\beta_t\tilde{\boldsymbol{\epsilon}}_\pm\mathrm{d}t + \frac{\sigma_t}{\alpha_t}\mathrm{d}\tilde{\boldsymbol{\epsilon}}_\pm \\
=& \big(\mathrm{d}\big(\frac{\sigma_t}{\alpha_t}\big) \mp \frac{\sigma_t}{\alpha_t}\beta_t\mathrm{d}t\big) \cdot \boldsymbol{\epsilon}_\phi(\cdots) \pm \frac{\sigma_t}{\alpha_t}\beta_t\tilde{\boldsymbol{\epsilon}}_\pm\mathrm{d}t \mp \frac{\sigma_t}{\alpha_t}\tilde{\boldsymbol{\epsilon}}_\pm\beta_t\mathrm{d}t + \sqrt{2\beta_t}\big(\frac{\sigma_t}{\alpha_t}\big)\mathrm{d}\boldsymbol{w}_t \\
=& \big(\mathrm{d}\big(\frac{\sigma_t}{\alpha_t}\big) \mp \frac{\sigma_t}{\alpha_t}\beta_t\mathrm{d}t\big) \cdot \boldsymbol{\epsilon}_\phi(\boldsymbol{x}'_\pm, t, y) + \sqrt{2\beta_t}\big(\frac{\sigma_t}{\alpha_t}\big)\mathrm{d}\boldsymbol{w}_t.
\end{aligned}
\tag{20}
$$

So $\boldsymbol{x}'_\pm$ and $\boldsymbol{x}_\pm$ follow the same dynamics. $\qquad\square$

We present a stochastic sampler in Algo. 3 that is equivalent Algorithm 2 in EDM (Karras et al., 2022) to show a practice implementation of Eq. 18 for sampling.

### G.3 PROPERTIES OF $\hat{\boldsymbol{x}}^{\mathrm{c}}_\pm$

### G.3.1 $\hat{\boldsymbol{x}}^{\mathrm{c}}_t$ ARE CLEAN IMAGES FOR ALL $t \in [t_s, T]$

**Lemma 1** (Sample predictions are non-noisy images). *The sample prediction of the diffusion model*

$$
\hat{\boldsymbol{x}}^{gt}_t \triangleq \frac{\boldsymbol{x}_t - \sigma_t\boldsymbol{\epsilon}_\phi(\boldsymbol{x}_t, t, y)}{\alpha_t}
\tag{21}
$$

*is a weighted average of images in the target distribution $p_0(\boldsymbol{x}_0)$:*

$$
\hat{\boldsymbol{x}}^{gt}_t = \mathbb{E}[\boldsymbol{x}_0|\boldsymbol{x}_t].
\tag{22}
$$

*Thus, $\hat{\boldsymbol{x}}^{gt}_t$ are non-noisy images. Furthermore,*

$$
\boldsymbol{\epsilon}_\phi(\boldsymbol{x}_t, t, y) = \frac{\boldsymbol{x}_t - \alpha_t\mathbb{E}[\boldsymbol{x}_0|\boldsymbol{x}_t]}{\sigma_t}.
\tag{23}
$$

*proof.*

$$\hat{\boldsymbol{x}}_t^{\text{gt}} = \frac{\boldsymbol{x}_t - \sigma_t \boldsymbol{\epsilon}_\phi(\boldsymbol{x}_t, t, y)}{\alpha_t}$$

$$= \frac{1}{\alpha_t} \big( \boldsymbol{x}_t + \sigma_t^2 \nabla_{\boldsymbol{x}_t} \log p_t(\boldsymbol{x}_t) \big)$$

$$= \frac{1}{\alpha_t} \big( \boldsymbol{x}_t + \frac{\sigma_t^2}{p_t(\boldsymbol{x}_t)} \nabla_{\boldsymbol{x}_t} p_t(\boldsymbol{x}_t) \big)$$

$$= \frac{1}{\alpha_t} \big( \boldsymbol{x}_t + \frac{\sigma_t^2}{p_t(\boldsymbol{x}_t)} \nabla_{\boldsymbol{x}_t} \int p(\boldsymbol{x}_t|\boldsymbol{x}_0) p_0(\boldsymbol{x}_0) \mathrm{d}\boldsymbol{x}_0 \big)$$

$$= \frac{1}{\alpha_t} \big( \boldsymbol{x}_t + \frac{\sigma_t^2}{p_t(\boldsymbol{x}_t)} \int p(\boldsymbol{x}_t|\boldsymbol{x}_0) \nabla_{\boldsymbol{x}_t} \log p(\boldsymbol{x}_t|\boldsymbol{x}_0) p_0(\boldsymbol{x}_0) \mathrm{d}\boldsymbol{x}_0 \big)$$

$$= \frac{1}{\alpha_t} \big( \boldsymbol{x}_t + \frac{\sigma_t^2}{p_t(\boldsymbol{x}_t)} \int p(\boldsymbol{x}_t|\boldsymbol{x}_0) \nabla_{\boldsymbol{x}_t} \big( - \frac{(\boldsymbol{x}_t - \alpha_t \boldsymbol{x}_0)^2}{2\sigma_t^2} \big) p_0(\boldsymbol{x}_0) \mathrm{d}\boldsymbol{x}_0 \big) \qquad (24)$$

$$= \frac{1}{\alpha_t} \big( \boldsymbol{x}_t - \frac{1}{p_t(\boldsymbol{x}_t)} \int p(\boldsymbol{x}_t|\boldsymbol{x}_0)(\boldsymbol{x}_t - \alpha_t \boldsymbol{x}_0) p_0(\boldsymbol{x}_0) \mathrm{d}\boldsymbol{x}_0 \big)$$

$$= \frac{1}{\alpha_t} \big( \boldsymbol{x}_t - \boldsymbol{x}_t \frac{\int p(\boldsymbol{x}_t|\boldsymbol{x}_0) p_0(\boldsymbol{x}_0) \mathrm{d}\boldsymbol{x}_0}{p_t(\boldsymbol{x}_t)} + \alpha_t \int \boldsymbol{x}_0 \frac{p(\boldsymbol{x}_t|\boldsymbol{x}_0) p_0(\boldsymbol{x}_0)}{p_t(\boldsymbol{x}_t)} \mathrm{d}\boldsymbol{x}_0 \big)$$

$$= \frac{1}{\alpha_t} \big( \boldsymbol{x}_t - \boldsymbol{x}_t + \alpha_t \int \boldsymbol{x}_0 p(\boldsymbol{x}_0|\boldsymbol{x}_t) \mathrm{d}\boldsymbol{x}_0 \big)$$

$$= \int \boldsymbol{x}_0 p(\boldsymbol{x}_0|\boldsymbol{x}_t) \mathrm{d}\boldsymbol{x}_0$$

$$= \mathbb{E}[\boldsymbol{x}_0|\boldsymbol{x}_t].$$

Thus,

$$\boldsymbol{\epsilon}_\phi(\boldsymbol{x}_t, t, y) = \frac{\boldsymbol{x}_t - \alpha_t \hat{\boldsymbol{x}}_t^{\text{gt}}}{\sigma_t} = \frac{\boldsymbol{x}_t - \alpha_t \mathbb{E}[\boldsymbol{x}_0|\boldsymbol{x}_t]}{\sigma_t}. \qquad (25)$$

$\square$

---

**Algorithm 3** A SDE sampler that is equivalent to Algorithm 2 in EDM (Karras et al., 2022)

1: **Input:** Diffusion model (sample prediction) $D_\phi$, $t_{i \in \{0, \cdots, N\}}$, $\gamma_{i \in \{0, \cdots, N-1\}}$, $S_{\text{noise}}$.
2: **Output:** $\hat{\boldsymbol{x}}_N^{\text{c}}$.
3: Initialize $\tilde{\boldsymbol{\epsilon}}_0 \sim \mathcal{N}(\boldsymbol{0}, \boldsymbol{I})$, $\hat{\boldsymbol{x}}_0^{\text{c}} = \boldsymbol{0}$
4: **for** $i \in \{0, \cdots, N-1\}$ **do**
5:      Sample $\boldsymbol{\epsilon}_i \sim \mathcal{N}(\boldsymbol{0}, S_{\text{noise}}^2 \boldsymbol{I})$
6:      $\hat{t}_i \leftarrow t_i + \gamma_i t_i$
7:      $\tilde{\boldsymbol{\epsilon}}_{i+1} \leftarrow \frac{t_i}{\hat{t}_i} \tilde{\boldsymbol{\epsilon}}_i + \sqrt{1 - (\frac{t_i}{\hat{t}_i})^2} \boldsymbol{\epsilon}_i$
8:      $\boldsymbol{d}_i \leftarrow (\hat{\boldsymbol{x}}_i^{\text{c}} - D_\phi(\hat{\boldsymbol{x}}_i^{\text{c}} + \hat{t}_i \tilde{\boldsymbol{\epsilon}}_{i+1}, \hat{t}_i))/\hat{t}_i$
9:      $\hat{\boldsymbol{x}}_{i+1}^{\text{c}} \leftarrow \hat{\boldsymbol{x}}_i^{\text{c}} + (t_{i+1} - \hat{t}_i) \boldsymbol{d}_i$
10:      **if** $t_{i+1} \neq 0$ **then**
11:         $\boldsymbol{d}_i' \leftarrow (\hat{\boldsymbol{x}}_{i+1}^{\text{c}} - D_\phi(\hat{\boldsymbol{x}}_{i+1}^{\text{c}} + t_{i+1} \tilde{\boldsymbol{\epsilon}}_{i+1}, t_{i+1}))/t_{i+1}$
12:         $\hat{\boldsymbol{x}}_{i+1}^{\text{c}} \leftarrow \hat{\boldsymbol{x}}_i^{\text{c}} + (t_{i+1} - \hat{t}_i)(\frac{1}{2}\boldsymbol{d}_i + \frac{1}{2}\boldsymbol{d}_i')$        $\triangleright$ Apply 2nd order correction
13:      **end if**
14: **end for**
15: Return $\hat{\boldsymbol{x}}_N^{\text{c}}$

---

**Proposition 2** ($\hat{\boldsymbol{x}}_\pm^{\text{c}}$ are non-noisy images). *$\hat{\boldsymbol{x}}_\pm^{\text{c}}$ in Eq. 18 are non-noisy images for all $t \in [t_s, T]$.*

*proof.* Since the initial conditions of $\hat{\boldsymbol{x}}_\pm^{\text{c}}$ ($\hat{\boldsymbol{x}}_-^{\text{c}} = \boldsymbol{0}$ for reverse process and $\hat{\boldsymbol{x}}_+^{\text{c}} \sim p_0(\boldsymbol{x}_0)$ for forward process) implies $\hat{\boldsymbol{x}}_\pm^{\text{c}}$ are initialized as non-noisy images, we only need to show that the dynamic of $\hat{\boldsymbol{x}}_\pm^{\text{c}}$ will not introduce Gaussian noise into $\hat{\boldsymbol{x}}_\pm^{\text{c}}$.

The dynamic of $\hat{x}_{\pm}^{c}$ can be reformulated as:

$$
\begin{aligned}
\mathrm{d}\hat{x}_{\pm}^{c} &= \big(\mathrm{d}\big(\frac{\sigma_t}{\alpha_t}\big) \mp \frac{\sigma_t}{\alpha_t}\beta_t\mathrm{d}t\big) \cdot \big(\epsilon_\phi(\alpha_t\hat{x}_{\pm}^{c} + \sigma_t\tilde{\epsilon}_{\pm}, t, y) - \tilde{\epsilon}_{\pm}\big), \\
&= \big(\mathrm{d}\big(\frac{\sigma_t}{\alpha_t}\big) \mp \frac{\sigma_t}{\alpha_t}\beta_t\mathrm{d}t\big) \cdot \frac{\alpha_t\hat{x}_{\pm}^{c} + \sigma_t\tilde{\epsilon}_{\pm} - \alpha_t\mathbb{E}[x_0|\alpha_t\hat{x}_{\pm}^{c} + \sigma_t\tilde{\epsilon}_{\pm}] - \sigma_t\tilde{\epsilon}_{\pm}}{\sigma_t}, \\
&= \big(\mathrm{d}\big(\log\frac{\sigma_t}{\alpha_t}\big) \mp \beta_t\mathrm{d}t\big) \cdot (\hat{x}_{\pm}^{c} - \mathbb{E}[x_0|\alpha_t\hat{x}_{\pm}^{c} + \sigma_t\tilde{\epsilon}_{\pm}]).
\end{aligned}
\tag{26}
$$

As Eq. 26 shows that $\hat{x}_{\pm}^{c}$ is always moving towards non-noisy sample prediction $\hat{x}_t^{\mathrm{gt}} = \mathbb{E}[x_0|x_t]$ for all $t \in [t_s, T]$, $\hat{x}_{\pm}^{c}$ will be non-noisy for all $t \in [t_s, T]$. □

We also visualize $\hat{x}_{\pm}^{c}$ at random timestep $t \in [0, T]$ of Stable Diffusion (Rombach et al., 2022) sampling processes in Appx. Fig. 17 to show that they are visually clean (non-noisy). We use clean variable to refer to $\hat{x}_{\pm}^{c}$ in this work since it is always non-noisy.

### G.3.2 INITIALIZATION OF $\hat{x}_t^{c}$

The initial condition of reverse-time clean flow SDE (Eq. 18) is given by $x_- = 0$ and $\tilde{\epsilon}_- \sim \mathcal{N}(0, I)$. This is consistent with a typical initialization of NeRF: the whole scene of the NeRF being all grey.

When we set $\hat{x}_-^{c} = 0$ as the initial condition for the clean flow SDE, it corresponds to the initial condition $x_- \sim \mathcal{N}(0, \sigma_T^2 I)$ (Karras et al., 2022) in the diffusion SDE (Eq. 17). However, since we set a small nonzero $\alpha_T$ at the beginning, the strict initial condition of the diffusion SDE should be $p_T(x_T)$, which is slightly different from $\mathcal{N}(0, \sigma_T^2 I)$. In this case, we should set $\hat{x}_-^{c} \sim p_0(x_0)$ in the clean flow SDE to make the initial condition of the two SDE identical. Prior works usually ignore the small difference between $p_T(x_T)$ and $\mathcal{N}(0, \sigma_T^2 I)$ and starts from pure noise when sampling (Lin et al., 2024), and from our practical observation, given different initial $\hat{x}_-^{c} \neq 0$ but the same $\tilde{\epsilon}_-$, clean flow SDE will yield almost identical outputs (given the same seeds), which implies the endpoints of $\hat{x}_t^{c}$ are not sensitive to initialization of $\hat{x}_t^{c}$. So we choose to set $\hat{x}_-^{c} = 0$ in this work as the initial condition.

### G.3.3 ENDPOINTS OF $\hat{x}_t^{c}$

At the end of the reverse-time clean flow SDE, $\hat{x}_-^{c} = x_0 \sim p_0(x_0)$. So $\hat{x}_t^{c}$ also ends as a sample in the target distribution $p_0(x_0)$ as $x_0$ in the reverse-time diffusion SDE.

### G.4 PROPERTIES OF $\tilde{\epsilon}_{\pm}$

$\tilde{\epsilon}_{\pm}$ can be seen as the "pure noise" part in the clean flow SDE (Eq. 18). Notably, the evolution of $\tilde{\epsilon}_{\pm}$ does not depend on $\hat{x}_t^{c}$ and has a closed-form solution. The dynamic of $\tilde{\epsilon}_{\pm}$ is given by

$$
\mathrm{d}\tilde{\epsilon}_{\pm} = \mp\tilde{\epsilon}_{\pm}\beta_t\mathrm{d}t + \sqrt{2\beta_t}\mathrm{d}w_t.
\tag{27}
$$

The initial condition for $\tilde{\epsilon}_{\pm}$ in both the forward and reverse process are $\tilde{\epsilon}_{\pm} \sim \mathcal{N}(0, I)$.

### G.4.1 CLOSED-FORM SOLUTIONS

For the forward process,

$$
\begin{aligned}
\mathrm{d}\big(e^{\int_0^t \beta_s\mathrm{d}s}\tilde{\epsilon}_+\big) &= e^{\int_0^t \beta_s\mathrm{d}s}\mathrm{d}\tilde{\epsilon}_+ + \tilde{\epsilon}_+\beta_t e^{\int_0^t \beta_s\mathrm{d}s}\mathrm{d}t \\
&= e^{\int_0^t \beta_s\mathrm{d}s}\sqrt{2\beta_t}\mathrm{d}w_t - \tilde{\epsilon}_+\beta_t e^{\int_0^t \beta_s\mathrm{d}s}\mathrm{d}t + \tilde{\epsilon}_+\beta_t e^{\int_0^t \beta_s\mathrm{d}s}\mathrm{d}t \\
&= e^{\int_0^t \beta_s\mathrm{d}s}\sqrt{2\beta_t}\mathrm{d}w_t.
\end{aligned}
\tag{28}
$$

Integral on both side of Eq. 28, we have

$$
e^{\int_0^t \beta_s\mathrm{d}s}\tilde{\epsilon}_+ - \tilde{\epsilon}_0 = \int_0^t \sqrt{2\beta_s}e^{\int_0^s \beta_r\mathrm{d}r}\mathrm{d}w_s.
\tag{29}
$$

Thus, we obtain the solution of $\tilde{\boldsymbol{\epsilon}}_+$:

$$\tilde{\boldsymbol{\epsilon}}_+ = e^{-\int_0^t \beta_s \mathrm{d}s} \cdot \tilde{\boldsymbol{\epsilon}}_0 + e^{-\int_0^t \beta_s \mathrm{d}s} \int_0^t \sqrt{2\beta_s} e^{\int_0^s \beta_r \mathrm{d}r} \mathrm{d}\boldsymbol{w}_s. \tag{30}$$

Similarly, we can obtain the solution of $\tilde{\boldsymbol{\epsilon}}_-$:

$$\tilde{\boldsymbol{\epsilon}}_- = e^{-\int_t^T \beta_s \mathrm{d}s} \cdot \tilde{\boldsymbol{\epsilon}}_T + e^{-\int_t^T \beta_s \mathrm{d}s} \int_T^t \sqrt{2\beta_s} e^{\int_s^T \beta_r \mathrm{d}r} \mathrm{d}\bar{\boldsymbol{w}}_s. \tag{31}$$

Specifically, we can derive a closed-form formulation to compute $\tilde{\boldsymbol{\epsilon}}_+(t)$ given $\tilde{\boldsymbol{\epsilon}}_+(t')$ for $t' < t$ from Eq. 30, which takes the following form:

$$\tilde{\boldsymbol{\epsilon}}_+(t) = e^{-\int_{t'}^t \beta_s \mathrm{d}s} \tilde{\boldsymbol{\epsilon}}_+(t') + \sqrt{1 - e^{-2\int_{t'}^t \beta_s \mathrm{d}s}} \boldsymbol{\epsilon}, \tag{32}$$

$$= \sqrt{1 - \gamma} \tilde{\boldsymbol{\epsilon}}_+(t') + \sqrt{\gamma} \boldsymbol{\epsilon}, \tag{33}$$

where

$$\gamma = 1 - e^{-2\int_{t'}^t \beta_s \mathrm{d}s}, \ \boldsymbol{\epsilon} \sim \mathcal{N}(\boldsymbol{0}, \boldsymbol{I}). \tag{34}$$

For $\tilde{\boldsymbol{\epsilon}}_-(t)$ and $t' > t$,

$$\tilde{\boldsymbol{\epsilon}}_-(t) = e^{-\int_t^{t'} \beta_s \mathrm{d}s} \tilde{\boldsymbol{\epsilon}}_-(t') + \sqrt{1 - e^{-2\int_t^{t'} \beta_s \mathrm{d}s}} \boldsymbol{\epsilon}, \tag{35}$$

$$= \sqrt{1 - \gamma} \tilde{\boldsymbol{\epsilon}}_-(t') + \sqrt{\gamma} \boldsymbol{\epsilon}, \tag{36}$$

where

$$\gamma = 1 - e^{-2\int_t^{t'} \beta_s \mathrm{d}s}, \ \boldsymbol{\epsilon} \sim \mathcal{N}(\boldsymbol{0}, \boldsymbol{I}). \tag{37}$$

### G.4.2 SPECIAL CASE SOLUTION OF DDPM

DDPM (Ho et al., 2020) corresponds to a special choice of $\beta_t$, where $\beta_t = \frac{\mathrm{d}(\sigma_t/\alpha_t)/\mathrm{d}t}{\sigma_t/\alpha_t}$ (Karras et al., 2022). We present the solution of Eq. 35 when $\beta_t$ corresponds to the choice of DDPM in the following:

$$\tilde{\boldsymbol{\epsilon}}_-(t) = \frac{\sigma_t/\alpha_t}{\sigma_T/\alpha_T} \tilde{\boldsymbol{\epsilon}}_T + \sqrt{1 - \left(\frac{\sigma_t/\alpha_t}{\sigma_T/\alpha_T}\right)^2} \boldsymbol{\epsilon}. \tag{38}$$

Assuming a designed schedule such that a $k$-step DDPM has a constant $\gamma$ in two consecutive steps as in Eq. 11. We get $\tilde{\boldsymbol{\epsilon}}_-(k) = (1 - \gamma)^{\frac{k}{2}} \tilde{\boldsymbol{\epsilon}}_-(0) + (1 - (1 - \gamma)^k)^{\frac{1}{2}} \boldsymbol{\epsilon}$. Thus, we obtain a value of $\gamma$ in Eq. 11 that corresponds to DDPM:

$$\gamma = 1 - \left(\frac{\sigma_t/\alpha_t}{\sigma_T/\alpha_T}\right)^{\frac{2}{k}} \approx \frac{2 \log \frac{\sigma_T/\alpha_T}{\sigma_t/\alpha_t}}{k}. \tag{39}$$

Putting a typical parameter configuration in our experiments with Stable Diffusion (DDPM sampler) into Eq. 39, where $t \approx 0.212$, $\sigma_t/\alpha_t \approx 0.60$, $T \approx 0.974$, $\sigma_T/\alpha_T \approx 12.59$ and $k = 25000$, we get $\gamma \approx 0.00024$.

### G.4.3 VARIANCE OF $\tilde{\boldsymbol{\epsilon}}_{\pm}$

All vector components of $\tilde{\boldsymbol{\epsilon}}_{\pm}$ are of unit variance for all $t \in [0, T]$:

$$\begin{aligned}
\mathrm{Var}(\tilde{\boldsymbol{\epsilon}}_{+,i}) &= e^{-2\int_0^t \beta_s \mathrm{d}s} + e^{-2\int_0^t \beta_s \mathrm{d}s} \int_0^t 2\beta_s e^{2\int_0^s \beta_r \mathrm{d}r} \mathrm{d}s \\
&= e^{-2\int_0^t \beta_s \mathrm{d}s}(1 + \int_0^t 2\beta_s e^{2\int_0^s \beta_r \mathrm{d}r} \mathrm{d}s) \\
&= e^{-2\int_0^t \beta_s \mathrm{d}s}(1 + \int_0^t \mathrm{d}e^{2\int_0^s \beta_r \mathrm{d}r}) \\
&= e^{-2\int_0^t \beta_s \mathrm{d}s}(1 + e^{2\int_0^t \beta_r \mathrm{d}r} - 1) \\
&= 1,
\end{aligned} \tag{40}$$

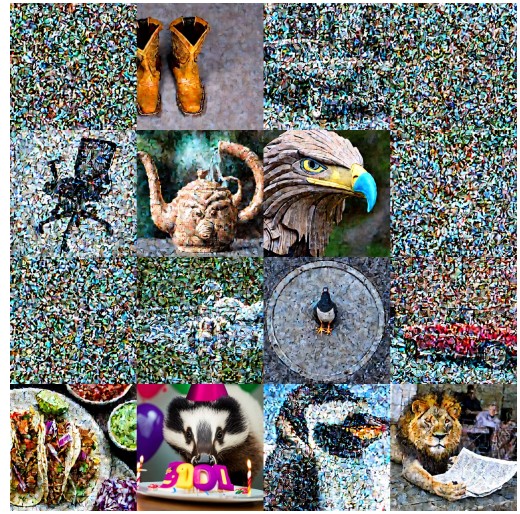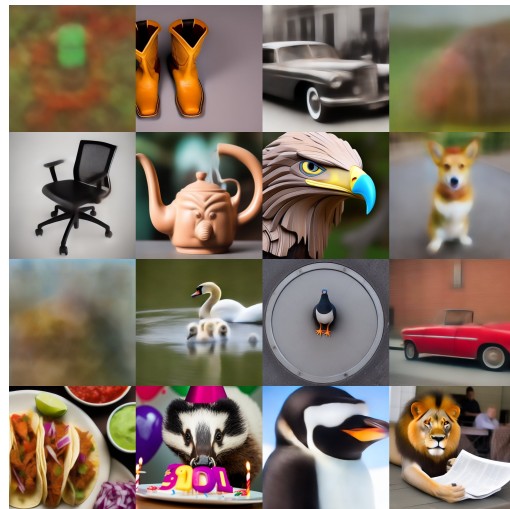

Figure 16: **Visualization of noisy variable $x_t$.**  Figure 17: **Visualization of clean variable $\hat{x}_t^c$.**

$$
\begin{aligned}
\mathrm{Var}(\tilde{\boldsymbol{\epsilon}}_{-,i}) &= e^{-2\int_t^T \beta_s \mathrm{d}s} + e^{-2\int_t^T \beta_s \mathrm{d}s} \int_t^T 2\beta_s e^{2\int_s^T \beta_r \mathrm{d}r} \mathrm{d}s \\
&= e^{-2\int_t^T \beta_s \mathrm{d}s}(1 + \int_t^T 2\beta_s e^{2\int_s^T \beta_r \mathrm{d}r} \mathrm{d}s) \\
&= e^{-2\int_t^T \beta_s \mathrm{d}s}(1 - \int_t^T \mathrm{d}e^{2\int_s^T \beta_r \mathrm{d}r}) \\
&= e^{-2\int_t^T \beta_s \mathrm{d}s}(1 - 1 + e^{2\int_t^T \beta_r \mathrm{d}r}) \\
&= 1.
\end{aligned}
\tag{41}
$$

### G.5 Clean Flow ODE

When we set $\beta_t = 0$ in clean flow SDE (Eq. 18), it becomes determined and changes to an ODE (Eq. 8). Furthermore, $\mathrm{d}\tilde{\epsilon}_{\pm} = 0$ and thus $\tilde{\epsilon}_{\pm}$ will become a constant $\tilde{\epsilon}$. This ODE is the same ODE presented in FSD (Yan et al., 2024). It is also equivalent to the signal-ODE presented in BOOT (Gu et al., 2023) when the diffusion model is changed to sample-prediction.

## H Discussion on the Choice of the Variable Space

### H.1 Ground-truth Variable

Apart from the clean variable $\hat{x}_t^c$, FSD (Yan et al., 2024) also defined another variable space that is visually clean, which is the *ground-truth variable* $\hat{x}_t^{gt}$. $\hat{x}_t^{gt}$ is defined by

$$
\hat{\boldsymbol{x}}_t^{gt} \triangleq \frac{\boldsymbol{x}_t - \sigma_t \boldsymbol{\epsilon}_\phi(\boldsymbol{x}_t, t, y)}{\alpha_t}.
\tag{42}
$$

$\hat{x}_t^{gt}$ is also known as the "sample prediction" of the diffusion model. The ODE on $\hat{x}_t^{gt}$ is given by:

$$
\mathrm{d}\hat{\boldsymbol{x}}_t^{gt} = -(\frac{\sigma_t}{\alpha_t}) \cdot \mathrm{d}\boldsymbol{\epsilon}_\phi(\boldsymbol{x}_t, t, y).
\tag{43}
$$

Concurrent work SDI (Lukoianov et al., 2024) shares an insight similar to ours by also using rendered images to replace the "non-noisy variables" to guide 3D generation. The difference between SDI (Lukoianov et al., 2024) and our method is that SDI replaced the ground-truth variable $\hat{x}_t^{gt}$ with rendered image $\boldsymbol{g}_\theta(\boldsymbol{c})$ but we replace the clean variable $\hat{x}_t^c$ with $\boldsymbol{g}_\theta(\boldsymbol{c})$.

Theoretically speaking, if it's just to solve the OOD problem when using image PF-ODE as a guidance for 3D generation, we think it's both reasonable to replace $\hat{x}_t^{gt}$ and $\hat{x}_t^c$ with rendered images,

since they are both non-noisy throughout the diffusion process (Lemma 1 and Proposition 2). However, it's difficult to exactly compute the update rule in Eq. 43 since $\boldsymbol{x}_t$ is required on right hand side of Eq. 43. In order to recover $\boldsymbol{x}_t$ given $\hat{\boldsymbol{x}}_t^{\text{gt}}$, SDI needs to solve a fixed point equation, which is hard to be solved (Lukoianov et al., 2024). In practice, SDI use a loss gradient similar to ISM. SDI interpret the DDIM inversion as the approximated solution of the fixed point equation. Difficulties also appear in works that attempt to apply guidance on the ground-truth variable $\hat{\boldsymbol{x}}_t^{\text{gt}}$ for conditional image generation, as seen in UGD (Bansal et al., 2023) and FreeDoM (Yu et al., 2023). In contrast, we can compute the evolution of $\hat{\boldsymbol{x}}_{\pm}^{\text{c}}$ exactly according to Eq. 18 without the need to solve a fixed point equation.

Additionally, another recent work ISM (Liang et al., 2023) can also be viewed as replacing the ground-truth variable $\hat{\boldsymbol{x}}_t^{\text{gt}}$ as discussed in SDI (Lukoianov et al., 2024), since the main difference between ISM and SDI loss is whether to apply text condition when computing DDIM inversion.

## H.2 COMPARISON WITH CONSISTENT3D

Consistent3D (Wu et al., 2024b) introduced the Consistency Distillation Sampling (CDS) loss by modifying the consistency training loss within the score distillation framework. Their insights into the connection between SDS and Diffusion SDE align closely with ours. However, their CDS loss stems from the consistency model training loss, similar to how SDS is derived from the diffusion model training loss, disregarding the Jacobian term (Poole et al., 2023). In contrast, our CFD loss directly follows the principles of diffusion model sampling through the ODE/SDE formulation. The image rendered from a specific camera view corresponds directly to a point on the ODE/SDE trajectory, resulting in distinct final training losses that differ from their CDS loss. Furthermore, our approach integrates a multiview consistent noising strategy, enhancing both consistency and robustness.

From a theoretical perspective, our work provides a more rigorous mathematical connection between score distillation and diffusion sampling compared with Consistent3D. Specifically: (i) While Consistent3D suggests that SDS can be interpreted as a form of SDE sampling, their proof relies on approximating the diffusion process by assuming optimal training at each step, an assumption that may not hold in practical experiments. In contrast, our approach does not rely on optimal training at every step. Additionally, our theory (Eq. 10 in our paper) covers a broader range of diffusion SDEs, including EDM (Karras et al., 2022) and PF-ODE as a special case. (ii) The CDS approach lacks a direct correspondence to a probability flow ODE trajectory, while our interpretation establishes a direct mapping between rendered images and points on the ODE/SDE trajectory.

## H.3 PROPERTIES OF CLEAN VARIABLE

Since clean flow ODE is a special case of clean flow SDE when $\beta_t = 0$, $\hat{\boldsymbol{x}}_t^{\text{gt}}$ in the ODE also maintains the "clean properties" discussed in Appx. G.3.

