# OpenReview forum: "Consistent Flow Distillation for Text-to-3D Generation"
_ICLR.cc/2025/Conference — ICLR 2025 Poster_

### Official Review · Reviewer_a4cX · 2024-10-30

**Soundness:** 3
**Presentation:** 2
**Contribution:** 2
**Rating:** 6
**Confidence:** 3

**Summary:**

The authors propose consistent flow distillation (CFD) strategy which can replace existing score distillation sampling (SDS) which leverages pre-trained 2D diffusion models for 3D generative models. The authors propose to guide 3D generation with 2D clean flow gradients operating jointly on a 3D object. They identify that a key in this process is to make the flow guidance consistent across different camera views.

**Strengths:**

1. The authors well compare other proposed SDS variants including VSD and ISM.
2. The author demonstrates the qualitative results of the proposed method compared to other variants.

**Weaknesses:**

1. The concern is that the proposed method leverages both MVDream and StableDiffusion2 as text-to-2D diffusion model, but other competitive methods, DreamFusion, ProlificDreamer, and LucidDreamer only StableDiffusion2. It means that the superiority of the proposed method might be from not the proposed CFD but from MVDream.

2. The proposed method only do experiments on text-to-3D tasks where the proposed CFD can be applied to any X-to-3D models.

**Questions:**

1. What happen if the proposed CFD is applied on DreamFusion pipeline which only replaces SDS to CFD while keeping all the other components the same?

2. What happen if the proposed CFD is applied to image-to-3D models like Wonder3D? Please show some results more than text-to-3D task.

3. Please give some results on user study for text-to-3D generative models where most of the recent text-to-3D generative models also show this results (or show SSIM or LPIPS results on image-to-3D tasks where there exist GT images on other camera views).

---

> ### Author Response · Authors · 2024-11-22
> **Response to Reviewer a4cX (1/2)**
>
> Thank you for the valuable feedback. We address your comments in the following. Please let us know if you have additional questions.
>
> ---
>
> **W1:** CFD leverages both MVDream and StableDiffusion2 as text-to-2D diffusion model but the baselines didn’t.
>
> **A1:**  We clarify that we do **NOT** use MVDream in all comparison experiments (Table 2, 3 and 5, and Figure 4, 12 and 13) to ensure fair comparisons to the baselines. MVDream is only used in Figure 1, 6-11, where the baselines also used MVDream in Figure 11. Importantly, our method without MVDream still outperforms the baselines, as demonstrated in Fig. 4, Fig. 13, and Tables 2, 3 and 5.
>
> ---
>
> **W2:** The proposed method only do experiments on text-to-3D tasks where the proposed CFD can be applied to any X-to-3D models.
>
> **A2:** We argue that this should not be considered as a weakness. Most prior works on score distillation are proposed for one task, typically text-to-3D generation. While their improved theory and practice are potentially useful in various other tasks (including image-to-3D, depth-to-3D, and one-step diffusion distillation), this is instead a strength of the generality of the method, and it is hard to have solid results on all tasks in one work. We detail the points below:
>
> 1. Most prior works on score distillation methods [1–5], including the recently published work [6], focus solely on evaluating text-to-3D tasks in their studies and do not explore other X-to-3D applications.
>
> 2. Text-to-3D tasks present unique challenges compared to image-to-3D due to their multimodal nature. The unpredictability of text-to-3D often leads to oversmoothing or a lack of diversity in score distillation results. However, our method successfully addresses these challenges, generating diverse outputs with high-quality textures, as shown in Fig. 1 and 6. These results have already demonstrated the effectiveness of our methods.
>
> 3. Extending to image-to-3D pipelines often requires additional design considerations, as many state-of-the-art synthesis models either lack support for flexible camera views [7–10], generate lower-quality backviews [11–12], or remain close-sourced [13]. To address these limitations, incorporating a refinement stage, as proposed in our paper, would likely be necessary to enhance performance. However, implementing such adaptations within the short rebuttal period is not feasible.
>
> ---
>
> [1] Wang, Zhengyi, et al. "Prolificdreamer: High-fidelity and diverse text-to-3d generation with variational score distillation." Advances in Neural Information Processing Systems 36 (2024).
>
> [2] Liang, Yixun, et al. "Luciddreamer: Towards high-fidelity text-to-3d generation via interval score matching." Proceedings of the IEEE/CVF Conference on Computer Vision and Pattern Recognition. 2024.
>
> [3] Wu, Zike, et al. "Consistent3d: Towards consistent high-fidelity text-to-3d generation with deterministic sampling prior." Proceedings of the IEEE/CVF Conference on Computer Vision and Pattern Recognition. 2024.
>
> [4] Chen, Rui, et al. "Fantasia3d: Disentangling geometry and appearance for high-quality text-to-3d content creation." Proceedings of the IEEE/CVF international conference on computer vision. 2023.
>
> [5] Huang, Yukun, et al. "Dreamtime: An improved optimization strategy for diffusion-guided 3d generation." The Twelfth International Conference on Learning Representations. 2023.
>
> [6] McAllister, David, et al. "Rethinking Score Distillation as a Bridge Between Image Distributions." arXiv preprint arXiv:2406.09417 (2024).
>
> [7] Wu, Kailu, et al. "Unique3D: High-Quality and Efficient 3D Mesh Generation from a Single Image." arXiv preprint arXiv:2405.20343 (2024).
>
> [8] Shi, Ruoxi, et al. "Zero123++: a single image to consistent multi-view diffusion base model." arXiv preprint arXiv:2310.15110 (2023).
>
> [9] Li, Peng, et al. "Era3D: High-Resolution Multiview Diffusion using Efficient Row-wise Attention." arXiv preprint arXiv:2405.11616 (2024).
>
> [10] Long, Xiaoxiao, et al. "Wonder3d: Single image to 3d using cross-domain diffusion." Proceedings of the IEEE/CVF Conference on Computer Vision and Pattern Recognition. 2024.
>
> [11] Liu, Ruoshi, et al. "Zero-1-to-3: Zero-shot one image to 3d object." Proceedings of the IEEE/CVF international conference on computer vision. 2023.
>
> [12] Voleti, Vikram, et al. "Sv3d: Novel multi-view synthesis and 3d generation from a single image using latent video diffusion." European Conference on Computer Vision. Springer, Cham, 2025.
>
> [13] Gao, Ruiqi, et al. "Cat3d: Create anything in 3d with multi-view diffusion models." arXiv preprint arXiv:2405.10314 (2024).

---

> ### Author Response · Authors · 2024-11-22
> **Response to Reviewer a4cX (2/2)**
>
> **Q1:** What happen if the proposed CFD is applied on DreamFusion pipeline which only replaces SDS to CFD while keeping all the other components the same?
>
> **A3:** The experiments are exactly conducted and detailed in the paper: Figures 4, 12, and 13 present the results of our method where both the baselines and CFD distill only SDv2.1 (without MVDream).
>
> ---
>
> **Q2:** What happen if the proposed CFD is applied to image-to-3D models like Wonder3D?
>
> **A4:** As discussed in rebuttal **A2**, the novel synthesis model in Wonder3D is constrained to six fixed camera positions, lacking support for flexible camera views. This limitation makes it unsuitable for direct use with score distillation, which requires the ability to query across a wider range of camera views to generate meshes effectively. However, score distillation can be employed in a second refinement stage to enhance texture quality, similar to the approach in our paper and as demonstrated in concurrent work such as DreamCraft3D++ [1].
>
> [1] Sun, Jingxiang, et al. "DreamCraft3D++: Efficient Hierarchical 3D Generation with Multi-Plane Reconstruction Model." arXiv preprint arXiv:2410.12928 (2024).
>
> ---
>
> **Q3:** Please give some results on user study for text-to-3D generative models where most of the recent text-to-3D generative models also show this results.
>
> **A5:** We conducted an additional user study involving 20 participants to evaluate the quality of our CFD-generated samples compared to official samples from baseline models. Each participant was presented with 8 sample pairs in random order and asked to select the best sample in each pair. The results of this study are as follows:
>
> | |Dreamfusion (SDS)|ProlificDreamer (VSD)|LucidDreamer (ISM)|
> |---|---|---|---|
> |Percentage Preference for CFD|95.0%|65.0%|73.3%|
>
> These results demonstrate the strong preference for our method over the baselines, highlighting its effectiveness and quality improvements.
>
>
> ---
>
> Please do not hesitate to let us know if you have any additional comments.

---

> ### Comment · Reviewer_a4cX · 2024-12-03
>
> I appreciate the authors’ response and there is no remaining question. I’ll raise my score to borderline accept

---

### Official Review · Reviewer_kcGa · 2024-11-03

**Soundness:** 2
**Presentation:** 3
**Contribution:** 2
**Rating:** 6
**Confidence:** 3

**Summary:**

This paper proposes Consistent Flow Distillation (CFD), which leverages gradients from 2D image flows to achieve better consistency across views, thereby improving 3D generation quality and diversity.

**Strengths:**

1. The generation quality of this proposed method is very high. The textures are realistic and detailed, providing a high level of visual fidelity that closely resembles real-world materials.
2. This paper is well-written, with well-organized sections that guide the reader through theory and methodology.

**Weaknesses:**

1. Concurrent work. I believe it is necessary for the authors to clarify the distinction between their approach and “Consistent Flow Distillation for Test-to-3D Generation” within the main body of this paper. The ODF-based optimization and multi-view consistent Gaussian noise used here are quite similar to those in FSD, which warrants a more explicit comparison.
2. Experimental Setup. (a) It’s unclear whether CFD utilizes MVDream in the comparison experiments, and if so, this may introduce an unfair advantage. (b) Only ten prompts are used in the comparative experiments. A broader evaluation set would improve the robustness of the evaluation. (c) FSD also should be included in the comparison for a more comprehensive evaluation. (d) The generation diversity hasn't been well evaluated. I also suggest that aesthetic evaluation metrics, e.g., LAION Aesthetics Predictor [1] and PickScore [2], can provide a more holistic assessment of the generated textures. (e) The ablation study lacks visualizations, which would help in understanding the impact of different components of the proposed method.

[1] https://laion.ai/blog/laion-aesthetics. [2] Pick-a-Pic: An Open Dataset of User Preferences for Text-to-Image Generation.

**Questions:**

See the weaknesses section. I would consider raising my socre if the author can address my concerns above.

---

> ### Author Response · Authors · 2024-11-22
> **Response to Reviewer kcGa**
>
> Thank you for the valuable feedback. We address your comments in the following. Please let us know if you have additional questions.
>
> ---
>
> **W1:** The ODE-based optimization and multi-view consistent Gaussian noise used here are quite similar to those in concurrent work FSD, which warrants a more explicit comparison.
>
> **A1:** We note that FSD is non peer-reviewed arXiv preprint in recent months. We provide a detailed comparison between our method and FSD in Appendix Section E.3 (due to the page limit of the main paper). In summary, our approach differs from FSD in the following key aspects:
>
> 1. **Theoretical Scope:** FSD’s theory is limited to ODE-based optimization, while our method encompasses a broader range of diffusion SDEs. Notably, FSD can be viewed as a special case of our method when $\gamma = 0$.
> 2. **Noising Strategy:** FSD employs a simple spherical noising technique that aligns noise on a sphere rather than on the object surface. We have observed that this approach often results in over-smoothed geometries and suboptimal surface quality (as demonstrated in Fig. 13). In contrast, our method introduces a more robust noise formulation that is better aligned with object geometry, leading to significantly better results.
>
> Additionally, we have included further numerical experiments (see also rebuttal **A2.3** or Tab. 3 in our paper) to demonstrate these differences and their impact on performance.
>
> ---
>
> **W2.1:** Whether CFD utilizes MVDream in the comparison experiments?
>
> **A2.1:**  We clarify that we do NOT use MVDream in all comparison experiments (Table 2, 3 and 5, and Figure 4, 12 and 13) to ensure fair comparisons to the baselines. MVDream is only used in Figure 1, 6-11, where the baselines also used MVDream in Figure 11.
>
> ---
>
> **W2.2:** Only ten prompts are used in the comparative experiments. A broader evaluation set would improve the robustness of the evaluation.
>
> **A2.2:** While we experiment with 10 prompts in Tab. 2, we generate 10 distinct samples for each prompt, resulting in a total of 100 samples. For the FID calculation, this translates to 50,000 generated images. Furthermore, in Tab. 5, we evaluate our method using 128 prompts, which provides a significantly broader evaluation set.
>
> ---
>
> **W2.3:** The generation diversity hasn't been well evaluated.  I also suggest that aesthetic evaluation metrics, e.g., LAION Aesthetics Predictor [1] and PickScore [2], can provide a more holistic assessment of the generated textures.
>
> **A2.3:** We note that FID can be influenced by diversity as it compares the distribution of deep features (while there might be no perfect metric yet for only evaluating diversity). Our FID measurement in Table 2 includes generated samples across different random seeds, ensuring that diversity is accounted for in our evaluation. Fig. 11 also visualized the diversity of our method and the baseline. Additionally, we have conducted new experiments incorporating the suggested aesthetic evaluation metrics, such as the LAION Aesthetics Predictor and PickScore, and have also included FSD for a comprehensive comparison. In this experiment, both the baselines and our CFD method distilled **only SDv2.1**. Following the evaluation protocol established in Diffusion-DPO [1], we report the score winning rates of CFD compared to baseline methods. Due to time and resource limitations, the evaluation was conducted on 50 randomly selected prompts from the DreamFusion dataset. The results are summarized below (and Tab. 3 in the revised version of the paper):
>
>
> |Method (win rate)|LAION score|Pick score|
> |---|---|---|
> |CFD vs. SDS|0.54|0.64|
> |CFD vs. VSD|0.60|0.68|
> |CFD vs. ISM|0.56|0.66|
> |CFD vs. FSD|0.54|0.78|
> |CFD vs. CDS|0.68|0.68|
>
> The results demonstrate that our CFD method consistently outperforms the baseline models, including FSD, across both LAION and Pick score metrics. This further highlights the robustness and effectiveness of CFD in generating high-quality results.
>
> [1] Wallace, Bram, et al. "Diffusion model alignment using direct preference optimization." Proceedings of the IEEE/CVF Conference on Computer Vision and Pattern Recognition. 2024.
>
> ---
>
> **W2.4:** The ablation study lacks visualizations, which would help in understanding the impact of different components of the proposed method.
>
> **A2.4:** Thank you for the suggestion. The ablation on the noise design and the flow space is visualized in Fig. 5, and ablation on components of our methods is in Fig. 15. We have also added additional ablation studies, including the impact of MVDream (Fig. 14), in the revised version of the paper. These include visualizations that illustrate the contribution of different components to the overall performance, providing a clearer understanding of their effects.
>
> ---
>
> Please do not hesitate to let us know if you have any additional comments.

---

> > ### Comment · Reviewer_kcGa · 2024-11-25
> >
> > I appreciate the authors for providing additional experimental results, which addressed my concerns regarding the experimental section. Considering the high-quality generation results demonstrated by this method, I have decided to raise my score. However, I still believe the paper should include a clear discussion in the main text about the differences between this work and concurrent work FSD, as well as prior work Consistent3D (as mentioned by Reviewer DP49).

---

> > > ### Author Response · Authors · 2024-11-28
> > > **Response to Reviewer kcGa**
> > >
> > > Thank you for your feedback. We have updated **A2.3** with a quantitative comparison to Consistent3D (CDS), which will be included in the final version. We will also include a clear discussion of the differences between our work, FSD, and CDS in the main text in our finial version.
> > >
> > > Thank you again for your helpful suggestions.

---

### Official Review · Reviewer_DP49 · 2024-11-04

**Soundness:** 3
**Presentation:** 2
**Contribution:** 2
**Rating:** 6
**Confidence:** 4

**Summary:**

This work proposes Consistent Flow Distillation (CFD) to enhance generation diversity and quality in SDS-based text-to-3D generation task. By treating the SDS as a trajectory of SDE, the authors propose guiding the optimization process via consistent 2D clean flow gradients. A key insight is maintaining consistent 2D image flows across different viewpoints for generating high-quality 3D outputs. To achieve this, they present an algorithm for computing multi-view consistent Gaussian noise, aligning noise textures precisely on the 3D object's surface. Extensive experiments showcase the effectiveness of CFD over related methods like VSD, ISM.

**Strengths:**

- The paper stands out for its robust presentation of results, thorough experimental analysis, and compelling evidence.
- Introducing Consistent Flow Distillation, the paper leverages 2D clean flow gradients and multi-view consistent noise to elevate the diversity and quality of 3D generation.
- Through empirical results, it is evident that the proposed CFD effectively enhances the diversity of generated outputs, showcasing its potency in improving the quality and variety of 3D-generated content.

**Weaknesses:**

- The stated contributions appear to overlap with existing methodologies.
    - The utilization of SDE formulations mirrors the approach outlined in "Consistent3D: Towards Consistent High-Fidelity Text-to-3D Generation with Deterministic Sampling Prior." While Consistent3D emphasizes addressing the unpredictability inherent in SDE sampling by introducing a deterministic sampling prior, the rationale behind employing image PF-ODE to steer 3D generation remains ambiguous.
    - The concept of multi-view consistent Gaussian noise is the same in "Geometry-Aware Score Distillation via 3D Consistent Noising and Gradient Consistency Modeling." Despite the advancements in quality seen in this work, a detailed comparative analysis is warranted. These approaches all seem to draw inspiration from Integral Noise.

- Introducing CFD could potentially inject more diversity into the 3D generation process. However, in SDS-based 3D generation, each iteration of inconsistent content distillation may exhibit the Janus problem. It remains uncertain whether CFD might improve multi-Janus issues, prompting the incorporation of MVDream for distillation. It could be beneficial to present results distilled from SDV2 or DeepFloy-IF to strengthen their argument.

**Questions:**

- The primary questions are raised in the Weaknesses part, which related to raising score.
- Could the paper provide a detailed comparison regarding memory usage and training time costs to existing methods?
- While MVDream is introduced to mitigate the Janus problem, there is a concern that it might overfit to the 3D training set, potentially resulting in object omission issues. Is there potential for CFD to address this drawback effectively? Typical prompts such as "A squirrel playing the guitar," "A pig wearing a backpack," and "a bear playing an electric bass" could shed light on this aspect.
- An open question: amidst various efforts to enhance SDS optimization for improved quality, can the authors assert that their formulation stands out as the best in Table 1?
- The significance of Figure 10 in the appendix, which likely demonstrates the effectiveness of CFD, suggests that its inclusion in the main body could boost the paper's impact and clarity.

---

> ### Author Response · Authors · 2024-11-22
> **Response to Reviewer DP49 (1/3)**
>
> Thank you for the valuable feedback. We address your comments in the following. Please let us know if you have additional questions.
>
> ---
> **W1.1:** The stated contributions seem to overlap with "Consistent3D: Towards Consistent High-Fidelity Text-to-3D Generation with Deterministic Sampling Prior.".
>
> **A1.1:** We note that the work Consistent3D (Wu et al.) is published in CVPR 2024 (June 17), which is within 4 months to the ICLR full-paper deadline and may be considered contemporaneous according to ICLR policy. We would like to clarify the distinction between our contributions and those of Consistent3D and will add it in our final version. While there may appear to be similarities, our contributions are fundamentally different in the following ways:
>
> 1. The Consistent3D authors propose the Consistency Distillation Sampling (CDS) loss by modifying the consistency training loss within the score distillation framework. However, their CDS loss originates from the consistency model training loss, analogous to how SDS can be derived from the diffusion model training loss while ignoring the Jacobian term [1]. In contrast, our CFD loss directly follows the principles of diffusion model sampling via ODE/SDE formulation.The image rendered from a specific camera view directly corresponds to a point on the ODE/SDE trajectory, leading to distinct final training losses that are not equivalent to their CDS loss. Additionally, our approach integrates a multiview consistent noising strategy, further enhancing the consistency and robustness of the method. Our perspective also explains the widely adopted timestep annealing technique [2, 3], whereas the training-based framework of Consistent3D only justifies random timestep sampling, similar to standard diffusion training. Quantitve comparison with CDS can be found in **Response to Reviewer kcGa, A2.3**.
>
> 2. From a theoretical standpoint, our work provides a more rigorous mathematical connection between score distillation and diffusion sampling compared to Consistent3D. Specifically:
>     - Consistent3D posits that SDS can be interpreted as a form of SDE sampling. However, their proof relies on approximating the diffusion process by assuming that each step is trained to optimality. This assumption may not consistently hold true in practical experiments. In contrast, our approach does not rely on the assumption of optimal training at every step. Additionally, our theory (Eq. 10 in our paper)  encompasses a broader range of diffusion SDEs in EDM [5], including PF-ODE as a special case.
>     - Their CDS approach lacks a direct correspondence to a probability flow ODE trajectory. In contrast, our interpretation establishes a direct mapping between rendered images and points on the ODE/SDE trajectory.
> ---
> **W1.2:**  Rationale behind employing image PF-ODE to steer 3D generation remains ambiguous.
>
> **A1.2:** The rationale for employing diffusion ODE/SDE to guide 3D generation stems from the fact that they are both the default diffusion sampling algorithms in current image generation pipelines. Using an ODE/SDE solver to perform diffusion sampling ensures precise adherence to the underlying probabilistic model, allowing for the generation of samples that align closely with the realistic image distribution.
>
> In the context of 3D generation, while score distillation methods like SDS and VSD also leverage diffusion to sample 3D objects, following diffusion ODE/SDE offers several advantages:
>
> 1. **Improved Interpretability:** PF-ODE/SDE provides a mathematically grounded framework that bridges the gap between image sampling and 3D generation, ensuring consistency across modalities.
> 2. **Theoretical Alignment:** Unlike other methods, PF-ODE/SDE directly aligns with the probabilistic foundations of diffusion models, offering a more principled approach to 3D sampling.
>
> As noted in [1, 6] and in the introduction of our paper, methods such as SDS or VSD are limited to generating samples near the maximum likelihood point of the distribution. This would result in lack of diversity and it is difficult for  their methods to sample from the whole valid distribution. However, employing PF-ODE/SDE allows for a more comprehensive exploration of the realistic distribution learned by the diffusion model, making it a robust and interpretable choice for steering 3D generation.
>
> ---
>
> [1] Poole, Ben, et al. "Dreamfusion: Text-to-3d using 2d diffusion."
>
> [2] Wang, Zhengyi, et al. "Prolificdreamer: High-fidelity and diverse text-to-3d generation with variational score distillation."
>
> [3] Huang, Yukun, et al. "Dreamtime: An improved optimization strategy for diffusion-guided 3d generation."
>
> [4] Wu, Zike, et al. "Consistent3d: Towards consistent high-fidelity text-to-3d generation with deterministic sampling prior."
>
> [5] Karras, Tero, et al. "Elucidating the design space of diffusion-based generative models."
>
> [6] Wang, Peihao, et al. "Taming mode collapse in score distillation for text-to-3d generation."

---

> ### Author Response · Authors · 2024-11-22
> **Response to Reviewer DP49 (2/3)**
>
> **W1.3:** The concept of multi-view consistent Gaussian noise is the same in "Geometry-Aware Score Distillation via 3D Consistent Noising and Gradient Consistency Modeling."
>
> **A1.3:** We note that the mentioned work GASD (Kwak et al.) is an arXiv preprint, also within 4 months to the ICLR full-paper deadline. We are happy to add the reference and discussions in our final version, and believe both works could be important contributions for text-to-3D generation.
>
> We would like to clarify that, while the concept of multi-view consistent Gaussian noise may appear similar, there are significant differences in implementation and approach:
>
> 1. **Our CFD Supports for More 3D Representations:** GASD introduces a multi-view consistent Gaussian noise technique specifically tailored to Gaussian splatting, as their method relies on point cloud upsampling. In contrast, our noise algorithm is designed to be agnostic to specific 3D representations. It supports a broad range of 3D formats, as long as they can render a depth map. This generalizability may allow for greater flexibility across different 3D pipelines.
>
> 2. **Different Noise Resampling Strategy:** The current version of the GASD paper does not clarify whether noise is resampled at each iteration or reused only within a batch. In contrast, our approach employs a mostly fixed noise throughout the generation process, ensuring better consistency and stability across iterations. This sampling strategy introduces significant theoretical differences compared to GASD. Additionally, as GASD has not yet released their code and omits several implementation details, it is challenging to perform a direct comparison of methodologies.
>
> ---
>
> **W2:** It remains uncertain whether CFD might improve multi-Janus issues, prompting the incorporation of MVDream for distillation. It could be beneficial to present results distilled from SDV2 or DeepFloy-IF to strengthen their argument.
>
> **A2:** Our work follows the research line of score distillation methods, including SDS, VSD, and ISM. The main focus of these works is not to improve the multi-Janus issue, and the theory of CFD is also not to address the multi-Janus issue. Resolving such issues is an orthogonal research direction, and we believe it may be more dependent on improving the teacher diffusion model rather than relying solely on score distillation methods.
>
> We clarify that we do **NOT** use MVDream in all comparison experiments (Table 2, 3 and 5, and Figure 4, 12 and 13) to ensure fair comparisons to the baselines. MVDream is only used in Figure 1, 6-11, where the baselines also used MVDream in Figure 11. We have included several results in our paper that are distilled exclusively from SDv2.1, as shown in Fig. 4, 12 and 13. These results demonstrate the performance of our method without additional shape enhancements. However, we observe that multi-face issues persist across all baselines and our method, particularly for prompts involving animals or humans, leading to a low success rate. This is why we incorporated MVDream into our complete pipeline. As demonstrated in the ablation study (Fig. 14) and discussed in Appendix Sec. E.1, MVDream serves as a shape initialization component in our pipeline, similar to how ISM utilizes Point-E for initializing their Gaussian splatting.
>
> ---
>
> **Q2:** What is the memory usage and training time of CFD and baseline?
>
> **A3:** We report the training time (on NVIDIA-L40 GPU) of baselines and our methods on the same prompt in the following table. The setting is the same experiment setting as in Tab. 2.
>
> | Method | SDS | VSD | ISM | CFD (ours) |
> |---|---|---|---|---|
> |Time (SDv2.1, 25000 iter)|1h19min|2h37min|1h59min|1h26min|
>
> The memory usage is fluctuating in our experiments due to NeRF pruning, but we did some optimization to fit all baselines and our CFD stably into 24GB memory of RTX-3090.

---

> ### Author Response · Authors · 2024-11-22
> **Response to Reviewer DP49 (3/3)**
>
> **Q3:** While MVDream is introduced to mitigate the Janus problem, there is a concern that it might overfit to the 3D training set, potentially resulting in object omission issues. Is there potential for CFD to address this drawback effectively? Typical prompts such as "A squirrel playing the guitar," "A pig wearing a backpack," and "a bear playing an electric bass" could shed light on this aspect.
>
> **A4:** It is possible that our CFD pipeline can regenerate missing objects during the refinement stage (Stage 2) when distilling SDv2.1. Since our complete pipeline comprises two stages and employs two different diffusion models, it benefits from the strengths of each. In particular, the second stage leverages SDv2.1, which is trained on a real-world dataset, enabling it to address object omission issues and refine results. Additionally, we tested the three prompts you proposed, which involve multiple objects: "A squirrel playing the guitar," "A pig wearing a backpack," and "A bear playing an electric bass." The results are available in the first section of our anonymous website: https://iclr25cfd.github.io/ and figure 8 in the revised version of our paper. CFD successfully generated all the objects in these scenarios, demonstrating its effectiveness in handling such complex prompts.
>
> ---
>
> **Q4:** An open question: amidst various efforts to enhance SDS optimization for improved quality, can the authors assert that their formulation stands out as the best in Table 1?
>
> **A5:** The short answer is yes. We believe that our method aligns most closely with the diffusion PF-ODE/SDE, offering significant theoretical advantages over alternative approaches. Since solving PF-ODE/SDE with an ODE/SDE solver forms the foundation of modern image generation pipelines, this alignment supports the effectiveness and robustness of our formulation. Nonetheless, based on the results presented in Table 2, 3 and 5, we believe that our approach currently stands out as the best.
>
> ---
>
> **Q5:** The significance of Figure 10 (11 in the new version) in the appendix, which likely demonstrates the effectiveness of CFD, suggests that its inclusion in the main body could boost the paper's impact and clarity.
>
> **A6:** Thank you for your suggestion. We agree that Figure 11 effectively highlights the performance of CFD and will try to move it into the main body in the final version of the paper. Currently, it is still in the appendix due to the page limit and the large size of the figure.
>
> ---
>
> Please do not hesitate to let us know if you have any additional comments.

---

> ### Author Response · Authors · 2024-11-28
> **Response to Reviewer DP49**
>
> Thank you for your thoughtful and constructive feedback. We greatly appreciate your suggestions, which have been valuable in improving the clarity and depth of our paper.
>
> As suggested, we will include a more detailed comparison with *Consistent3D* and the theory of SJC in the final version. We will also highlight the advantages of our method in aligning with the diffusion PF-ODE/SDE distribution, which we believe will provide a clearer understanding of our contributions.
>
> Regarding your comment on the initialization of $\hat{x}$, we would like to clarify that while the density is normal-initialized, the color of both the object and background are zero-initialized in our code base. As a result, the rendered images at initial steps are all gray, which is consistent with the zero initialization. We appreciate your attention to this detail and thank you again for pointing it out.
>
> Thank you again for your helpful suggestions.

---

### Official Review · Reviewer_vQut · 2024-11-09

**Soundness:** 4
**Presentation:** 3
**Contribution:** 2
**Rating:** 8
**Confidence:** 3

**Summary:**

This paper proposes the 'Consistent Flow Distillation (CFD)' for text-to-3D generation. It extends the success of SDE into 3D domain, and with its novel multi-view consistent Gaussian noise sampling, it demonstrates a simple yet effective ways to enhance the visual quality and diversity in 3D generation. Extensive quantitative and qualitative comparisons demonstrates its effectiveness compared to previous methods.

**Strengths:**

[**Novelty**]
- The adaptation of probability flow ODE (PF-ODE) with clean flow gradient from 2D images to guide 3D generation is innovative
- The adaptation of multi-view consistent Gaussian noise ensures a unified appearance from all angles, which is the key to high-fidelity texture generation

[**Significance**]
- The propose design of multi-view consistent noise is useful for the whole community, its performance boost in 3D-FID and 3D-CLIP scores compared to SDS, ISM, and VSD baselines, and exhibits richer details and more photorealistic textures, providing a considerable improvement to text-to-3D generation.

[**Completeness & Clarity**]
- I like its various qualitative comparisons with different baselines and ablations, also its examples of diverse generation for the same prompt.
- It is well-organized, and effectively explains advanced technical concepts, including clean flow gradients, PF-ODE, and the SDE-based sampling process.

**Weaknesses:**

- On significance and novelty, I think based on the current progress in the field of 3DGen AI, although CFD introduces innovative noise techniques, it doesn’t propose entirely new model architectures or evaluation metrics beyond standard score distillation approaches. This is more fundamental concern when existing 3D-generative models can generate high-quality 3D assets within minutes, which this approach can still take hours.
- Limited Qualitative Examples for long and complex Prompts: although the paper includes various qualitative comparisons, additional examples, especially for complex prompts, could further enhance understanding of CFD’s limitations
- I also feel that some example results are not sharp enough, like around line 820

**Questions:**

NA

---

> ### Author Response · Authors · 2024-11-22
> **Response to Reviewer vQut**
>
> Thank you for the valuable feedback. We address your comments in the following. Please let us know if you have additional questions.
>
> ---
>
> **W1.1:** Although CFD introduces innovative noise techniques, it doesn’t propose entirely new model architectures or evaluation metrics beyond standard score distillation approaches.
>
> **A1.1:** We highlight that our main contribution is not only the innovative noise techniques, but more importantly, they are based on the novel perspective and solid mathematical interpretation we proposed for existing score distillation techniques, which is non-trivial. These contributions together enable the generation of high-quality content while advancing the theoretical understanding of score distillation methods.
>
> ---
>
> **W1.2:** This is more fundamental concern when existing 3D-generative models can generate high-quality 3D assets within minutes, which this approach can still take hours.
>
> **A1.2:** Despite that the per-instance optimization paradigm is not optimal in runtime, the score distillation methods still remain an active [1-3] and valuable research direction due to their versatility and potential applications, including the applications that require runtime efficiency. For instance, a few minutes of refinement using score distillation can significantly enhance the texture quality of 3D assets in existing generation pipelines, as demonstrated by recent work like DreamCraft3D++ [4]. Moreover, score distillation techniques initially proposed for text-to-3D have proven effective in other domains, such as distilling faster image diffusion models [5, 6].
>
> [1] McAllister, David, et al. "Rethinking Score Distillation as a Bridge Between Image Distributions." arXiv preprint arXiv:2406.09417 (2024).
>
> [2] Liang, Yixun, et al. "Luciddreamer: Towards high-fidelity text-to-3d generation via interval score matching." Proceedings of the IEEE/CVF Conference on Computer Vision and Pattern Recognition. 2024.
>
> [3] Wu, Zike, et al. "Consistent3d: Towards consistent high-fidelity text-to-3d generation with deterministic sampling prior." Proceedings of the IEEE/CVF Conference on Computer Vision and Pattern Recognition. 2024.
>
> [4] Sun, Jingxiang, et al. "DreamCraft3D++: Efficient Hierarchical 3D Generation with Multi-Plane Reconstruction Model." arXiv preprint arXiv:2410.12928 (2024).
>
> [5] Sauer, Axel, et al. "Adversarial diffusion distillation." European Conference on Computer Vision. Springer, Cham, 2025.
>
> [6] Yin, Tianwei, et al. "One-step diffusion with distribution matching distillation." Proceedings of the IEEE/CVF Conference on Computer Vision and Pattern Recognition. 2024.
>
> ---
>
> **W2:** Limited Qualitative Examples for long and complex Prompts, additional examples, especially for complex prompts, could further enhance understanding of CFD’s limitations.
>
> **A2:** We have added new examples with long and complex prompts in the first section of our anonymous website: https://iclr25cfd.github.io/ and figure 8 in our revised paper. These qualitative examples demonstrate that our CFD performs well on such prompts, often matching the performance of the teacher diffusion model (SDv2.1). However, it is important to note that the limitations of our approach are influenced by the teacher model's ability to effectively interpret and respond to complex prompts.
>
> ---
>
> **W3:** I also feel that some example results are not sharp enough, like around line 820.
>
> **A3:** The perceived sharpness of some example results is primarily constrained by the performance of the teacher diffusion model (which also applies to other score distillation methods, including SDS, VSD). Additionally, the NeRF rendering process may inherently limit the representation of high-frequency details. Prior work [1] suggests that incorporating higher-resolution rendering, larger batch sizes, or leveraging diffusion models supporting higher resolutions could enhance visual fidelity. However, these approaches would significantly increase memory requirements and computational costs, and are orthogonal to the focus of this work.
>
> [1] Wang, Zhengyi, et al. "Prolificdreamer: High-fidelity and diverse text-to-3d generation with variational score distillation." Advances in Neural Information Processing Systems 36 (2024).
>
> ---
>
> Please do not hesitate to let us know if you have any additional comments.

---

### Author Response · Authors · 2024-11-22

We thank all reviewers for the feedback. We appreciate the reviewers recognizing the significance of our work, including a simple method based on novel theory (vQut), very high-quality results compared to prior works (vQut, DP49, kcGa), and thorough experimental analysis (vQut, DP49, a4cX).

We would like to highlight and clarify a main question from reviewers’ feedback:
- **Q: Is MVDream used in comparisons and lead to unfair advantage?**

  A: No, we do **NOT** use MVDream in all comparison experiments (Table 2, 3 and 5, and Figure 4, 12 and 13) to ensure fair comparisons to the baselines. MVDream is only used in Figure 1, 6-11, where the baselines also used MVDream in Figure 11. We thank the reviewers for their valuable feedback and have addressed this point in the revised version of our paper.

**Summary of revisions:** We summarize changes to our manuscript below; these changes have also been highlighted (red) in the new version. Updates on the anonymous website: https://iclr25cfd.github.io/ are also highlighted (red).
- Add additional samples with complex prompts in the paper appendix Fig. 8. Corresponding videos are also updated in the first section of the anonymous webpage.
- Include an additional ablation study figure on the usage of MVDream in appendix Sec. E.1. and Fig. 14.
- Update the experiment Tab. 3 in the main body of the paper and include additional aesthetic evaluation metrics required by reviewer kcGa.

Again, we thank the reviewers for their constructive feedback. We believe that all comments have been addressed in this revision, and are happy to address any further comments from reviewers.

Best,
Authors of CFD (submission 5833)

---

### Meta-Review · Area_Chair_gHXL · 2024-12-21

**Metareview:**

This paper presented a novel methodology for improving 3D consistency. Initially, the reviewers expressed concerns regarding the need for additional results, for instance with other baseline, and computational complexity analysis. However, after the rebuttal, they were satisfied with the supplementary results provided. All reviewers acknowledged the simplicity and straightforwardness of the method. The AC also reviewed the paper, the feedback, and the rebuttal, and similarly recognized the method as both simple and effective. Therefore, the AC recommends acceptance. The paper would benefit from including additional results and discussions in the final version.

**Additional Comments On Reviewer Discussion:**

After the rebuttal, they were satisfied with the supplementary results provided. All reviewers acknowledged the simplicity and straightforwardness of the method.

---

### Decision · Program_Chairs · 2025-01-22

Accept (Poster)